

# Four-quark scatterings in QCD II

**Wei-jie Fu[1,2], Chuang Huang[1]⋆, Jan M. Pawlowski[3,4] and Yang-yang Tan[1]**

**1** School of Physics, Dalian University of Technology, Dalian, 116024, P.R. China
**2** Shanghai Research Center for Theoretical Nuclear Physics,
NSFC and Fudan University, Shanghai 200438, China
**3** Institut für Theoretische Physik, Universität Heidelberg,
Philosophenweg 16, 69120 Heidelberg, Germany
**4** ExtreMe Matter Institute EMMI, GSI, Planckstraße 1, D-64291 Darmstadt, Germany

⋆ huangchuang@dlut.edu.cn

## Abstract

In [1], we initiated a program for the quantitative investigation of dynamical chiral symmetry breaking and resonant bound states in QCD with the functional renormalisation group, concentrating on the full infrared dynamics of four-quark scatterings. In the present work we extend this study and take into account a three-momentum channel approximation ($s, t, u$-channel) for the Fierz-complete four-quark vertices. We find that the four-quark vertex in this approximation is quantitatively reliable. In particular, we have computed the pion pole mass, pion decay constant, Bethe-Salpeter amplitudes, the quark mass function and wave function. Our results confirm previous findings that low energy effective theories only reproduce QCD quantitatively, if initiated with a relatively low ultraviolet cutoff scale of the order of 500 MeV. The quantitative description set up here paves the way for reliable quantitative access to the resonance structure in QCD within the fRG approach to QCD.

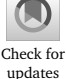

# 1  Introduction

The description of the formation process of hadron resonances and scattering processes as well as general timelike processes such as transport properties are intricate open problems in QCD. Recent years have seen significant progress in studies of first-principles QCD and hadron structure within both functional QCD and lattice QCD, for respective reviews see e.g. [2, 3] (functional QCD) and [4, 5] (lattice QCD).

Notably, functional QCD provides simple and direct computational access to the chiral limit and in particular, to timelike observables, see e.g. [6–9]. The latter observables are specifically relevant e.g., for scattering processes, a reliable description of the formation of resonances, as well as transport properties of QCD. These advantageous properties come with the price that the full system of coupled diagrammatic equations for correlation functions has to be truncated for numerical applications. This calls for systematic truncation schemes, whose systematic error can be measured and minimised with an apparent convergence of higher orders of the truncation scheme.

The present work is the second in a series of works, initiated in [1], where such a systematic expansion scheme within the functional renormalisation group (fRG) approach in QCD is developed, aiming at a quantitative description of hadron resonances and in particular their formation and scattering processes. This builds on and is complementary to previous fRG work in QCD [10–16]. In these works the focus was directed more toward studies of the phase structure of QCD and the respective vacuum works were embedded in this endeavor, for recent reviews see [3, 17].

In the present work we concentrate more on the comprehensive, Fierz-complete, description of the four-quark scattering vertices, including also timelike momenta. Specifically, this extends the Fierz-complete two-flavour [11, 13] to timelike momenta. Moreover, these works use emergent composites or dynamical hadronisation for the scalar-pseudoscalar channel, see [11, 15, 16, 18–21]: The fRG approach with emergent composites in the scalar-pseudoscalar channel includes the dynamics and multi-scattering processes of pions, which are the backbone of chiral perturbation theory ($\chi$PT). Accordingly, emergent composites will be included in the present approach at a later stage.

In the first paper in the series, [1], a single momentum channel dependence for all tensor structures of the four-quark vertex has been used in an NJL-type low energy effective theory (LEFT) [22–26]. The results indicated that quantitative access to the infrared dynamics of QCD requires at least a complete $s, t, u$-channel approximation, where $s, t, u$ are the Mandelstam variables. This step is done in the present work, where we aim for a quantitative description of the scattering physics of quarks and hadrons at low energies, both at spacelike and timelike momenta. While still working in an NJL-type LEFT, we also include part of the ultraviolet dynamics of QCD in terms of a QCD-assisted low energy effective theory [3, 17]. This systematic improvement allows an in-detailed study of the mechanisms at work and the evaluation of minimal approximation orders required for quantitative precision.

This work is organised as follows: In Section 2 we discuss the set-up of the QCD-assisted LEFT, including symmetric and antisymmetric four-quark dressings of different tensor structures, three-momentum-channel truncation for the four-quark vertices, etc. In Section 3, the physical scales of the LEFT are determined by the ratios of the physical pion mass with the pion decay constant in the chiral limit and at the physical point. Then, we compute the quark wave function and the mass function as well as the dressings of the Fierz-complete four-quark tensors and that of their crossing-symmetric partners. The results are used to compute observables such as the pion decay constant as a function of the pion mass, and the Bethe-Salpeter amplitudes of pion and $\sigma$-mode. We conclude in Section 4. The Appendices contain some technical details, plots and further comparisons.

## 2 QCD-assisted low energy effective theories

In [1] we have set up the quark-sector of low energy QCD with the Fierz complete four-quark scattering vertex. The Fierz completeness refers to a complete basis of momentum-independent tensor structures that should dominate the infrared dynamics in terms of a low momentum expansion. The momentum-dependent dressings were evaluated in a $t$-channel approximation. In the present work we improve on this approximation in two ways, that already prepare the stage for the full functional computation in first principles QCD that will be presented in [27].

We consider the purely fermionic effective action of low energy two-flavour QCD with a given infrared cutoff scale $k$,

$$\Gamma_k[q,\bar{q}] = \Gamma_{\text{kin},k}[q,\bar{q}] + \Gamma_{4q,k}[q,\bar{q}], \tag{1}$$

with the kinetic term

$$\Gamma_{\text{kin},k} = \int_p Z_q(p)\,\bar{q}(-p)\Big[i\slashed{p} + M_q(p)\Big]q(p), \tag{2}$$

with $\slashed{p} = \gamma_\mu p_\mu$. The quark wave function $Z_q$ and quark mass $M_q$ are $k$-dependent, and we have suppressed the respective argument. With (2), the full quark propagator is given by

$$G_q(p) = \frac{1}{Z_q(p)}\frac{-i\slashed{p}\left[1 + r_q\left(\frac{q^2}{k^2}\right)\right] + M_q(p)}{p^2\left[1 + r_q\left(\frac{q^2}{k^2}\right)\right]^2 + M_q^2(p)}, \tag{3}$$

with the shape function $r_q$ of the quark regulator function,

$$R_q(p) = Z_q(p)\,i\slashed{p}\,r_q\left(\frac{q^2}{k^2}\right). \tag{4}$$

In the limit of $k \to 0$ the propagator (3) reduces to

$$G_q(p) = \frac{1}{Z_q(p)}\frac{-i\slashed{p} + M_q(p)}{p^2 + M_q^2(p)}. \tag{5}$$

In (2) we have used the notation

$$\int_p \equiv \int \frac{d^4p}{(2\pi)^4}. \tag{6}$$

The second term in the effective action (1) is the full four-quark term in the effective action,

$$\Gamma_{4q,k} = -\int \frac{d^4p_1}{(2\pi)^4}\cdots\frac{d^4p_4}{(2\pi)^4}(2\pi)^4\delta(p_1 + p_2 + p_3 + p_4)$$

$$\times \lambda_\alpha(\boldsymbol{p})\mathcal{T}^{(\alpha)}_{ijlm}(\boldsymbol{p})\,\bar{q}_i(p_1)q_j(p_2)\bar{q}_l(p_3)q_m(p_4), \tag{7}$$

with

$$\boldsymbol{p} = (p_1, p_2, p_3, p_4).\tag{8}$$

In (7), a sum over $\alpha$ is implied, and the full set of tensors $\mathcal{T}^{(\alpha)}(\boldsymbol{p})$ is comprised of 512 tensors, see e.g. [9]. The indices $i, j, l, m$ include Lorentz, Dirac, flavour and color indices. The tensors $\mathcal{T}^{(\alpha)}$ can be ordered according to their momentum dependences in powers of $p_i$. A basis of the lowest momentum-independent order includes ten elements $\mathcal{T}^\alpha$, and we choose the set of tensors

$$\alpha \in \left\{ \sigma, \pi, a, \eta, (V \pm A), (V - A)^{\text{adj}}, \quad (S \pm P)^{\text{adj}}_-, (S + P)^{\text{adj}}_+ \right\},\tag{9}$$

see e.g. [11, 13, 28–32]. The parametrisation (7) and the set (9) have been already discussed and used in [1], for more details we refer the reader to this work.

A basis of the momentum-independent Fierz complete tensors with full crossing symmetry can be obtained as the symmetric component of the 10 tensors $\mathcal{T}^{(\alpha)}_{ijlm}$ listed in Appendix A,

$$\mathcal{T}^{(\alpha^-)}_{ijlm} = \frac{1}{2}\left( \mathcal{T}^{(\alpha)}_{ijlm} - \mathcal{T}^{(\alpha)}_{ljim} \right),\tag{10}$$

where the minus sign reflects the antisymmetry of the attached quark and anti-quark fields. The respective dressings $\lambda^+_\alpha(\boldsymbol{p})$ are positive under the commutation of momenta. The momentum-independent Fierz-complete basis $\{\mathcal{T}^{(\alpha^-)}\}$ in (10) can be accompanied by the antisymmetric part of $\mathcal{T}$,

$$\mathcal{T}^{(\alpha^+)}_{ijlm} = \frac{1}{2}\left( \mathcal{T}^{(\alpha)}_{ijlm} + \mathcal{T}^{(\alpha)}_{ljim} \right),\tag{11}$$

for more details see Appendix A. Seemingly, this is yet another set of momentum-independent tensor structures. However, their dressings are antisymmetric in $p_1 \leftrightarrow p_3$ and hence define tensor elements in the $p_i^2$-order basis. We emphasise that while the momentum-independent basis $\{\mathcal{T}^{(\alpha^+)}_{ijlm}\}$ in (10) is complete, the set $\{\mathcal{T}^{(\alpha^+)}_{ijlm}\}$ does not define an invariant sub-space in the $p_i^2$ tensor structures. Still, together with the momentum-independent tensors (10) they carry the crossing symmetry of the tensors $\mathcal{T}$.

In [1] we have computed the dressings $\lambda^+_\alpha$ of the Fierz complete tensors $\{\mathcal{T}^{(\alpha^-)}\}$ in the $t$-channel approximation, starting at a UV cutoff $\Lambda \approx 1\,\text{GeV}$. In the present work we improve the approximation used in [1] significantly in two directions:

Firstly, we include all Mandelstam channels $t, s, u$, (19), in our flows. Moreover, we accompany the momentum-independent tensor structures in (10) with the $p^2$-tensor structures in (11), as they are related by crossing symmetry. This improved approximation allows us to extend the computation deep into the chiral limit as well as largely reducing the systematic error.

Secondly, we use an initial condition at the UV cutoff scale in the spirit of QCD-assisted low energy effective theories: The current low energy effective theory emerges from first principles QCD due to a successive decoupling of the fundamental high energy degrees of freedom, specifically the gluon. This is discussed in detail in [3, 17]. As a consequence the quark two-point function and the four-quark vertex at the initial cutoff scale are given by the result of the ultraviolet fRG flow of QCD, in contradistinction to the classical input in an NJL-type model, This is discussed in Section 2.3.

## 2.1 Four-quark scattering vertex

The four-quark scattering vertex is given by four quark–anti-quark derivatives of the effective action,

$$\Gamma^{(4)}_{\bar{q}_i q_j \bar{q}_l q_m}(\boldsymbol{p}) = \frac{\delta^4 \Gamma_k[q, \bar{q}]}{\delta \bar{q}_i(p_1)\delta q_j(p_2)\delta \bar{q}_l(p_3)\delta q_m(p_4)},\tag{12}$$

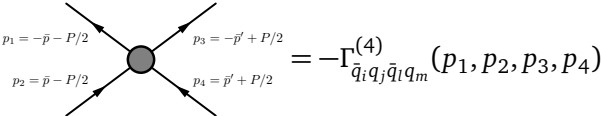

Figure 1: Diagrammatic representation of the quark four-point function. All momenta are counted as incoming and $i, j, l, m$ carry the Dirac, flavour and color indices. The notation for general 1PI $n$-point functions is depicted in Figure 15 in Appendix B.

with $\boldsymbol{p} = (p_1, ..., p_4)$ defined in (8). Applying (12) to (7), we are led to

$$\Gamma^{(4)}_{\bar{q}_i q_j \bar{q}_l q_m}(\boldsymbol{p}) = -2(2\pi)^4 \delta(p_1 + p_2 + p_3 + p_4) \times \sum_\alpha \left[ \lambda_\alpha(\boldsymbol{p}) \mathcal{T}^{(\alpha)}_{ijlm} - \lambda_\alpha(\boldsymbol{p}') \mathcal{T}^{(\alpha)}_{ljim} \right], \qquad (13)$$

with $\boldsymbol{p}' = (p_3, p_2, p_1, p_4)$. The combination in the last line encodes the crossing symmetry of the vertex. Equation (13) is depicted in Figure 1, where all momenta $p_1, ..., p_4$ are counted as incoming. Note that the blobs in our diagrams stand for the one-particle-irreducible (1PI) vertices, see Figure 15. This includes the 1PI two-point function or inverse propagator, see e.g. Figure 14. The full propagators are denoted by (inner) lines.

We expect that the contributions of the higher order tensors are suppressed in the flow due to the efficient momentum ordering in terms of $q^2/k^2 \lesssim 1$. Here, $q$ is the loop momentum which is cut off at $q^2 \approx k^2$. This suppression is very well tested by now: the fundamental diagrams that generate the four-quark vertex in the first place, carry the external momentum dependence in terms of propagators with a mass scale $\gtrsim k$, so higher order momentum dependences are suppressed. At its root this structure is also behind the great successes of the derivative expansion in the fRG approach, see [3]. However, in the present system with resonant interaction channels the derivative expansion is not only done in $p^2/k^2$ but also in terms of $p^2/m^2_{\text{res}}$. This has to be contrasted with QCD computations with dynamical hadronisation or emergent composites, as done in [11, 13] in two-flavour QCD. There, also the resonant channels are regularised.

Accordingly, the present approach requires a better momentum resolution of the resonant channels $\mathcal{T}^{(\sigma, \pi)}_{ijlm}$ as $p^2/m^2_{\text{res}}$ can be large and even diverges in the chiral limit for non-vanishing momentum.

In summary, we consider the influence of the higher order scattering terms with six or more quarks and anti-quarks as subleading. Indeed, they are strongly suppressed by phase space arguments: In short the suppression arguments rest upon the fact, that the form factor of such a scattering decays rapidly with the distance, and hence higher order scatterings are statistically suppressed. This suppression can only be violated in the presence of resonant interaction channels and diverging interactions, see e.g. the reviews [3, 17] for a more detailed discussion.

The dressings $\lambda^\pm(\boldsymbol{p})$ of the tensor structures (10) and (11) are derived from the dressings $\lambda^{(\alpha)}$ and the details are deferred to Appendix A. We are led to

$$\Gamma^{(4)}_{\bar{q}_i q_j \bar{q}_l q_m}(\boldsymbol{p}) = -4(2\pi)^4 \delta(p_1 + p_2 + p_3 + p_4) \times \sum_\alpha \left[ \lambda^+_\alpha(\boldsymbol{p}) \mathcal{T}^{(\alpha-)}_{ijlm} + \lambda^-_\alpha(\boldsymbol{p}) \mathcal{T}^{(\alpha+)}_{ijlm} \right], \qquad (14)$$

where $\lambda^\pm_\alpha(\boldsymbol{p})$ have the properties shown in (A.9) and (A.11) in Appendix A under permutations of momenta.

Finally, we remark that the dressings $\lambda(\boldsymbol{p})$ in (14) are not renormalisation group (RG) invariant and hence cannot be related directly to observables. Their RG-invariant counterparts

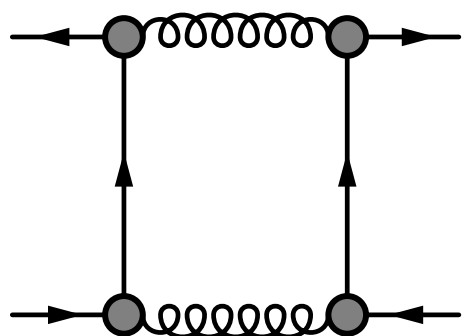

Figure 2: Schematic form of the quark-gluon box diagrams that contribute to the flow of the four-quark vertex in QCD, see e.g. [15]. The notation for general 1PI $n$-point functions is depicted in Figure 15 in Appendix B. The lines stand for full momentum-dependent propagators. Moreover, for the sake of simplicity we have dropped the cutoff insertion.

are given by

$$\bar{\lambda}(\boldsymbol{p}) = \frac{\lambda(\boldsymbol{p})}{\prod_{i=1}^{4} Z_q^{1/2}(p_i)}, \tag{15}$$

which are the dressings of the RG-invariant vertex $\bar{\Gamma}^{(4)}$ defined by

$$\Gamma^{(4)}_{\bar{q}_i q_j \bar{q}_l q_m}(\boldsymbol{p}) = \prod_{i=1}^{4} Z_q^{1/2}(p_i)\, \bar{\Gamma}^{(4)}_{\bar{q}_i q_j \bar{q}_l q_m}(\boldsymbol{p}), \tag{16}$$

and analogously for $\Gamma^{(n)}$. An expansion of the effective action in terms of the RG-invariant vertices $\bar{\Gamma}^{(n)}$ and the RG-invariant propagators is more rapidly converging in a given expansion scheme. This originates in the milder momentum dependence of the RG-invariant vertices, for a discussion see e.g. [13] in QCD and [33] in gravity. Note that the RG-invariant quark propagator $\bar{G}_q(p)$ follows from (2) with a multiplication $Z_q(p)$,

$$\bar{G}_q(p) = \frac{-\mathrm{i}\slashed{p}\left[1 + r_q\!\left(\frac{q^2}{k^2}\right)\right] + M_q(p)}{p^2\left[1 + r_q\!\left(\frac{q^2}{k^2}\right)\right]^2 + M_q^2(p)}, \tag{17}$$

and the RG invariance of (17) is manifest. In the limit of $k \to 0$ the RG-invariant propagator reduces to

$$\bar{G}_q(p) = \frac{-\mathrm{i}\slashed{p} + M_q(p)}{p^2 + M_q^2(p)}, \tag{18}$$

with the RG-invariant mass function $M_q(p)$.

## 2.2 $s, t, u$-channel truncation

In this work we use a $s, t, u$-channel approximation for the four-quark vertices as we have discussed in the introduction of the present Section. The Mandelstam variables read

$$s = (\bar{p} + \bar{p}')^2, \qquad t = P^2, \qquad u = (\bar{p} - \bar{p}')^2, \tag{19}$$

where the convention for momentum labels of the four-quark vertex is provided in Figure 1, and

$$P = -(p_1 + p_2), \qquad \bar{p} = \frac{p_2 - p_1}{2}, \qquad \bar{p}' = \frac{p_4 - p_3}{2}. \tag{20}$$

In the $s, t, u$-channel approximation for the four-quark vertices, the full momentum dependence of the four-quark dressings in (14) is approximated by

$$\lambda_\alpha^\pm(p_1, p_2, p_3, p_4) \approx \lambda_\alpha^\pm(s, t, u). \tag{21}$$

Put differently, we use the representation

$$\lambda_\alpha^\pm(p_1, p_2, p_3, p_4) = \lambda_\alpha^\pm(s, t, u) + \Delta\lambda_\alpha^\pm(p_1, p_2, p_3, p_4), \tag{22}$$

and assume

$$\Delta\lambda_\alpha^\pm(p_1, p_2, p_3, p_4) \approx 0. \tag{23}$$

The self-consistency of this approximation can be checked by computing $\Delta\lambda_\alpha^\pm$ on the results. In Appendix D we show that the size of $\Delta\lambda_\pi^\pm$ does not exceed 1.5% for the large range of configurations.

This approximation is qualitatively better than the single channel approximation, and we expect significantly improved results in comparison to [1]. In particular, we expect that the chiral limit is better accessible. This improvement comes at the price, that the numerical costs are higher. However, the $s, t, u$-channel approximation is far less costly than taking into account the full momentum dependence as shown in the left side of (21).

The configuration of four-momenta $P$, $\bar{p}$, and $\bar{p}'$ are chosen such that their four components in a Euclidean frame are given by

$$
\begin{aligned}
P_\mu &= \sqrt{P^2}\left(1, 0, 0, 0\right), \\
\bar{p}_\mu &= \sqrt{p^2}\left(1, 0, 0, 0\right), \\
\bar{p}'_\mu &= \sqrt{p^2}\left(\cos\theta, \sin\theta, 0, 0\right),
\end{aligned} \tag{24}
$$

with $\theta \in [0, \pi]$. Evidently, momenta $\bar{p}$ and $P$ are in the same direction, and $\bar{p}'$ has the same magnitude as $\bar{p}$ and the angle between them is given by $\theta$. Substituting (24) into (19), one arrives at

$$s = 2p^2(1 + \cos\theta), \qquad t = P^2, \qquad u = 2p^2(1 - \cos\theta). \tag{25}$$

It is readily verified that the subspace of the full momentum of four-quark vertices chosen here, $\{\sqrt{P^2}, \sqrt{p^2}, \cos\theta\}$, is in one-by-one correspondence to the subspace $\{s, t, u\}$. Note that this choice of the subspace is convenient for the investigation of properties of bound states in the following but not unique. Consequently, the split (22) depends on the choice of the configuration. We show in Appendix D, that the respective $\Delta\lambda_\pi^\pm$, that accounts for the deviation from the choice (24), is very small even in comparison to $\Delta\lambda$ for configurations with all channel momenta $t, s, u \neq 0$.

We conclude that $\Delta\lambda_\alpha^\pm$ accounts for subleading effects. Moreover, these effects are even averaged out when fed back to the diagrams. Hence, we consider our approximation trustworthy. A complete check of the systematics will be published elsewhere.

## 2.3  QCD-assisted initial conditions

It is left to discuss the initial condition for the effective action. For the effective action (1) this amounts to fixing the dressing functions

$$Z_q(p), \qquad M_q(p), \qquad \{\lambda_\alpha(\boldsymbol{p})\}, \tag{26}$$

at the initial scale $k = \Lambda \approx 1$ GeV. Naively one would put the initial values to 'classical' ones: the wave function of the quark is put to unity, and the (constituent) quark mass function is put

to the current quark value. Moreover, all vertex dressings of the four-quark tensor structures are set to zero, except the leading scalar-pseudo-scalar one, $\lambda_{\sigma,\Lambda} = \lambda_{\pi,\Lambda} = \lambda$. This amounts to setting up an NJL-type model and solving it in an elaborate approximation.

In the present work we take a first but significant step towards first principles QCD. There, the quark-gluon box diagrams dominate the flow of the four-quark scattering vertex for momenta $p^2 \gtrsim 1\,\text{GeV}$, and this class of diagrams is depicted schematically in Figure 2. The respective flow generates in particular symmetric scalar and pseudoscalar four-quark couplings $\bar{\lambda}_\sigma^+$ and $\bar{\lambda}_\pi^+$ with a non-vanishing momentum-dependence at the initial scale $k = \Lambda$, the latter being of the order of 1 GeV. Qualitatively, we can approximate these momentum-dependent initial conditions by

$$\lambda_{\sigma,\Lambda}^+(\bar{p}) = \lambda_{\pi,\Lambda}^+(\bar{p}) = \tilde{\lambda}\,\Lambda^2 \exp\left[-\frac{1}{2a}\left(\frac{\bar{p}}{\Lambda}\right)^2\right], \tag{27}$$

with the dimensionless parameter $\tilde{\lambda}$ and

$$\bar{p} = \frac{\sqrt{p_1^2 + p_2^2 + p_3^2 + p_4^2}}{2} = \frac{\sqrt{s+t+u}}{2}, \tag{28}$$

while all other four-quark couplings are set to zero. The two parameters in (27) can either be taken from the results in functional QCD, or they are fixed from phenomenological constraints.

## 3 Numerical results

With the set-up described in Section 2, we compute several low energy observables in QCD such as the the quark mass function and quark wave function, the pion mass, decay constant and Bethe-Salpeter amplitudes of the pion.

### 3.1 Determination of the physics scales

Any functional and lattice approach, both effective field theory ones and first principles ones, is defined on the level of the initial effective action or classical lattice action, and the scales are measured naturally in the respective momentum cutoff scale $k = \Lambda$ or $1/a$, where $a$ is the lattice spacing. Then the physics at hand or rather the mass scale has to be determined by fixing the input parameters with non-vanishing mass dimension.

In QCD these are the current quark masses while the strong coupling is not adjusted. The current masses of light $u$ and $d$ quarks are fixed by the value of the pole mass of the pion, $m_\pi \approx 138\,\text{MeV}$. As the current quark masses are the only explicit scale in QCD, $m_\pi$ has to be measured in a dynamical scale of QCD and typically the pion decay constant $f_\pi \approx 93\,\text{MeV}$ is taken. Hence we adjust $m_\pi/f_\pi \approx 138/93$. This requires the computation of the pole mass $m_\pi$ from the pole position and the pion decay constant from the Bethe-Salpeter (BS) amplitude $h_\alpha$ on the pole.

Both can be read off from the four-quark coupling $\lambda_\pi$ in the pseudoscalar channel: The pole mass is conveniently determined from the zero of the inverse four-quark coupling $\lambda_\pi(t)$ in the $t$-channel as the coupling diverges on the pion pole. As in [1] the extrapolation of momenta from the Euclidean to Minkowski regime is done using Padé approximants: While Padé approximants are not a good basis for reconstructions, they (and any other method) work well for the reconstruction of the location of the first singularity in the complex frequency plane. The relevant results of this extrapolation are shown in Figure 3 in physical units.

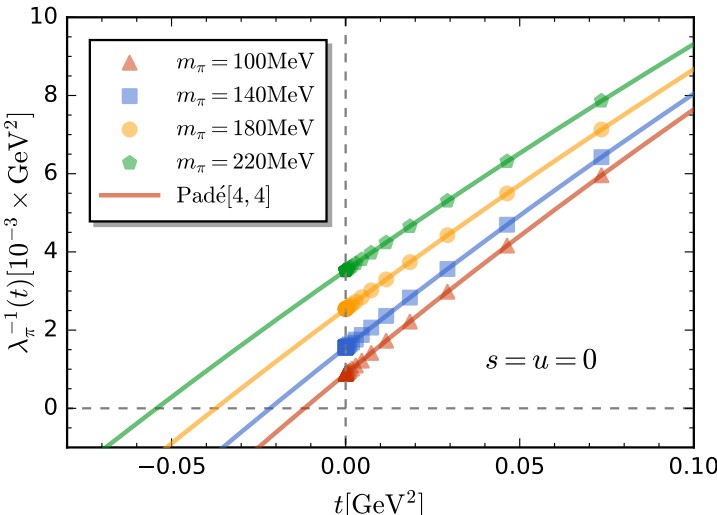

Figure 3: Inverse four-quark coupling in the $\pi$ channel, $1/\lambda_\pi(t)$, as a function of the Mandelstam variable $t = P^2$ with vanishing $s$ and $u$. Results for several different values of $m_\pi$ are compared. Data points denote the results calculated in the flow equation in the Euclidean $t > 0$ region. The solid lines stand for results of analytic continuation from $t > 0$ to $t < 0$ based on the fit of the Padé[4,4] approximants.

The pion decay constant can be computed from the BS amplitude $\bar{h}_\pi$ which is the amplitude of the $t$-channel pole part of the RG invariant four-quark scattering couplings $\bar{\lambda}_\pi$,

$$\bar{\lambda}_\pi(t = P^2 \to -m_\pi^2) = \frac{\bar{h}_\pi \bar{h}_\pi^\dagger}{P^2 + m_\pi^2} + \text{finite}. \tag{29}$$

Resolving the above relation for $\bar{h}_\alpha(p)$ with $\alpha = \sigma, \pi$ for the $u$-channel momentum $p$ leads us to

$$\bar{h}_\alpha(p) = \lim_{t \to -m_\alpha^2} \left[ \bar{\lambda}_\alpha(s = 0, t, u = 4p^2) \bar{G}_\alpha^{-1}(t) \right]^{1/2}, \tag{30}$$

with the RG-invariant normalised propagator of bound states $\bar{G}_\alpha(t) = 1/(t + m_\alpha^2)$. We note that in the QCD approach with emergent composites as used in [13, 15] the BS amplitude is nothing but the (RG-invariant) Yukawa coupling.

In Figure 4 we plot the BS amplitude as a function of the $u$-channel momentum $p$ on the pole $t = -m_\alpha^2$. Finally, the pion decay constant is defined as

$$\langle 0 | J_{5\mu}^a(x) | \pi^b \rangle = i P_\mu f_\pi \delta^{ab}, \tag{31}$$

where the left hand side reads

$$\langle 0 | J_{5\mu}^a(x) | \pi^b \rangle = \int \frac{d^4 q}{(2\pi)^4} \text{Tr} \left[ \gamma_\mu \gamma_5 T^a \bar{G}_q(q + P) \bar{h}_\pi(q) \gamma_5 T^b \bar{G}_q(q) \right], \tag{32}$$

with the RG-invariant propagator $\bar{G}_q$ in (17) and the BSE amplitude $\bar{h}_\pi$ in (30). Accordingly, (32) is manifestly RG-invariant. Finally, $T^a$ in (32) denotes the generators of the flavor $SU(N_f)$ group with normalisation

$$\text{tr } T^a T^b = \frac{1}{2} \delta_{ab}. \tag{33}$$

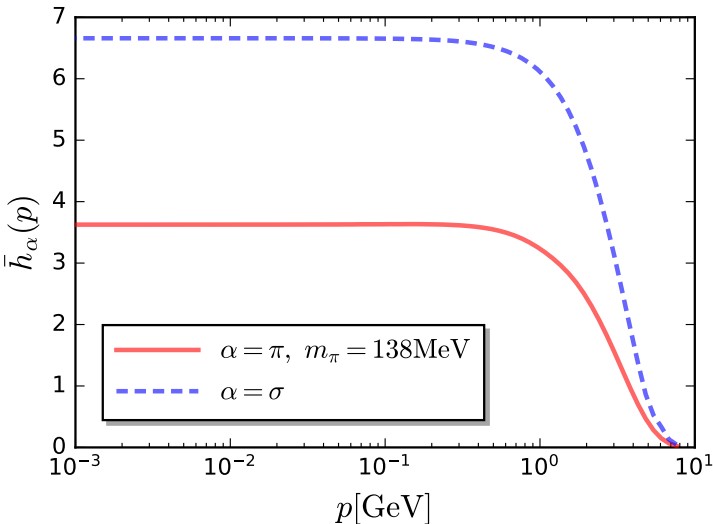

Figure 4: Bethe-Salpeter amplitudes $\bar{h}_\alpha$ of the pion (red line) and $\sigma$-mode (blue line) as functions of the momentum.

Equation (32) is computed on the pole of pion with $P^2 = -m_\pi^2$ which is the smallest mass scale in QCD. Thus, one can expand (32) in powers of $P$ and the leading term provides us with the leading-order expression of the pion decay constant,

$$f_\pi = 2N_c \int \frac{d^4q}{(2\pi)^4} \frac{\bar{h}_\pi(q)M_q(q)}{\left[q^2 + M_q^2(q)\right]^2} \,, \tag{34}$$

where $N_c = 3$ is the number of colors. Equation (34) is manifestly RG-invariant.

Adjusting the ratio $m_\pi/f_\pi \approx 138/93$ fixes the physical point and further observables, including those for other values of current quark masses, that are predictions. An example of the latter is the ratio of the pion decay constant in the chiral limit, $f_\pi^\chi$, and that at the physical point, i.e., $f_\pi^\chi/f_\pi = 86/93$. However, in a low energy effective theory this is not a prediction and only holds true if the low energy dynamics of QCD are emulated well. This entails that one has to fix effective low energy couplings that in QCD are generated from the fundamental quark-gluon dynamics.

The coupling parameters in the present low energy effective theory approach are the light current quark mass $m_l = m_u = m_d$ in the isospin-symmetric limit, the strength of the four-quark coupling $\tilde{\lambda}$ in the scalar-pseudoscalar tensor channel at the initial scale, and the parameter $a$ governing its momentum dependence as shown in (27). The strength parameter $\tilde{\lambda}$ and the momentum shape parameter $a$ are adjusted with a physics observable and hence accommodate the missing gluonic fluctuations with cutoff scales $k \leq \Lambda$ as well as truncation artifacts in the effective action.

This suggests the following procedure for determining the input parameters $m_l$, $\tilde{\lambda}$ and $a$ at the initial cutoff scale $k = \Lambda$: We use the ratios of pion pole mass $m_\pi$ with the pion decay constant $f_\pi$ at the physical point and with the pion decay constant $f_\pi^\chi$ in the chiral limit, i.e.,

$$\frac{m_\pi}{f_\pi} \approx \frac{138}{93} \,, \qquad \frac{m_\pi}{f_\pi^\chi} \approx \frac{138}{86} \,, \tag{35}$$

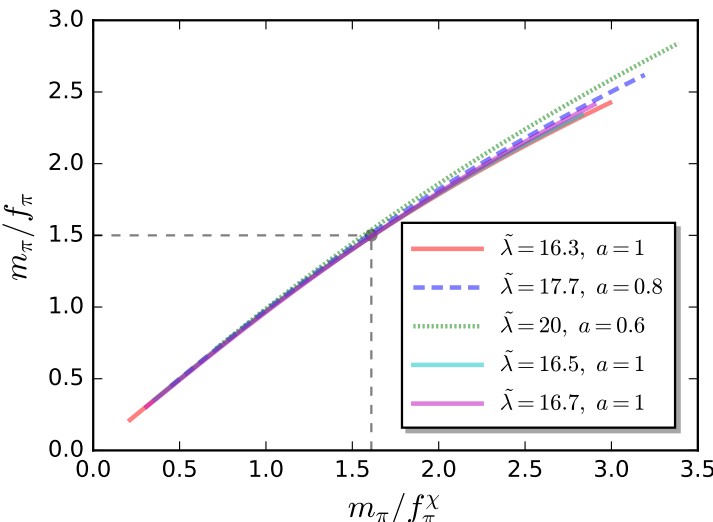

Figure 5: Ratio of the pion mass and the pion decay constant, $m_\pi/f_\pi$, as a function of the ratio of the pion mass and the pion decay constant in the chiral limit, $m_\pi/f_\pi^\chi$. Several sets of values of parameters $\tilde{\lambda}$ and $a$ for the four-quark coupling at the initial cutoff in (27) are utilised. The physical values $m_\pi^{\text{phys}}/f_\pi^{\text{phys}}$ and $m_\pi^{\text{phys}}/f_\pi^\chi$ are depicted as horizontal and vertical dashed lines respectively. The physical curve is obtained with $\tilde{\lambda} = 16.7$ and $a = 1$.

where the latter can also be written as $f_\pi/f_\pi^\chi \approx 93/86$. Roughly speaking, the first ratio is used to adjust the current quark mass $m_l$, and in first principles QCD the second would be a prediction. In the low energy effective theory the second one is used to calibrate the interaction of the low energy dynamics described by $\tilde{\lambda}$ and $a$ in (27). In Figure 5 we plot $m_\pi/f_\pi$ as a function of $m_\pi/f_\pi^\chi$, and the two physical values are given by the horizontal and vertical dashed line, respectively. The curve depends on the parameters $\tilde{\lambda}$ and $a$, and we select that which includes the physical point. This procedure fixes the parameters in the input coupling (27),

$$\bar{\lambda} = 16.7\,, \qquad a = 1\,. \tag{36}$$

We use the pion decay constant in the chiral limit, for fixing the absolute momentum scale as (34) works best there. We set $f_\pi^\chi \approx 86$ MeV and the ratios (35) then fix the physics scales at the physical point. In particular this fixes the physics scale of the initial cutoff scale $\Lambda$ and the initial light current quark mass,

$$\Lambda = 850\,\text{MeV}\,, \qquad m_l = 19.5\,\text{MeV}\,. \tag{37}$$

At lower cutoff scales in effective theories without gluon dynamics, the initial current quark mass here is larger than the current quark mass in QCD calculations. Furthermore, at the physical point we find the $\sigma$ meson mass $m_\sigma = 570$ MeV.

## 3.2 Observables

A first prediction in the present approach is the pion mass dependence of the pion decay constant for all $m_\pi$, depicted in Figure 6 in physical units in comparison to the result of chiral perturbation theory ($\chi$PT) at the order of $\mathcal{O}(p^4)$ [34], see also [35,36]. Our results are in excellent agreement with $\chi$PT for $m_\pi \lesssim 170$ MeV. For larger pion masses, $\chi$PT starts deviating from our results.

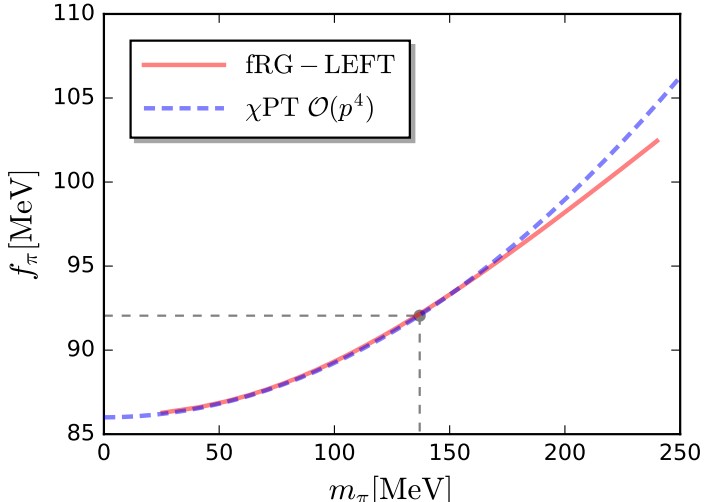

Figure 6: Pion decay constant as a function of the pion mass (tuned by the current quark mass) for the physical four-quark coupling for the initial condition (36) and $\Lambda = 850\,\text{MeV}$, see (37). We also show the result of chiral perturbation theory at the order of $\mathcal{O}(p^4)$ [34]. The gray dashed lines correspond to the physical point.

In Figure 7 we show the light constituent quark mass function at the physical pion mass in comparison to lattice results, [37], and the functional QCD results from [13] (fRG), see also [38–41] (DSE). For the ultraviolet cutoff scale of 850 MeV in (37), the decay of the quark mass function (solid red line) is far weaker than in QCD. This was to be expected, as LEFTs only describe the full dynamics of QCD for far lower momentum scales, see in particular [14, 42]. In a forthcoming paper, [27], we have embedded the quark sector in full QCD and the results there confirm that in [14, 42] in the present set-up.

Alternatively, we can lower the ultraviolet cutoff scale $\Lambda$ in (37) such, that the LEFT accommodates the full QCD dynamics for momenta below $\Lambda$. Such a case is also shown in Figure 7 (dashed blue lines) for the initial condition

$$\Lambda = 500\,\text{MeV}, \qquad m_l = 19.5\,\text{MeV}, \qquad \tilde{\lambda} = 20.4, \qquad a = 1. \tag{38}$$

As expected, the quark mass function shows a reasonable agreement with the QCD results. This entails both, a non-trivial reliability check of the current approximation as well as emphasising the relevance of gluon degrees of freedom in the momentum regime 500 - 1000 MeV. As also discussed in [14, 42], this relatively low quantitative UV cutoff scale limits the quantitative predictivity of LEFTs for larger external parameters such as temperature and quark chemical potential.

We proceed with the discussion of the results of the four-quark dressings $\bar{\lambda}_\alpha$. As discussed in [1] and in Section 2, we consider a set of twenty tensors derived from the set of tensors $\mathcal{T}^{(\alpha)}$ in (A.1): the Fierz-complete ten momentum-independent tensors $\mathcal{T}^{(\alpha^-)}$ defined in (10) that follow with crossing symmetry from the tensors in (A.1) and its symmetric part (11) with an antisymmetric dressing, that also follows from (A.1). We emphasise again that this part implicitly depends on the extension of the given $\mathcal{T}^{(\alpha^-)}$ with a symmetric part, and the set $\mathcal{T}^{(\alpha)}$ implies a specific choice. Their dressings are obtained in terms of their symmetric and antisymmetric parts,

$$\bar{\lambda}_\alpha(\boldsymbol{p}) = \bar{\lambda}_\alpha^+(\boldsymbol{p}) + \bar{\lambda}_\alpha^-(\boldsymbol{p}), \tag{39}$$

with $\lambda_\alpha^\pm$ defined in (A.8). While $\lambda_\alpha^+$ are the dressings of a Fierz complete basis, the $\lambda_\alpha^-$ reflect our choice (A.1).

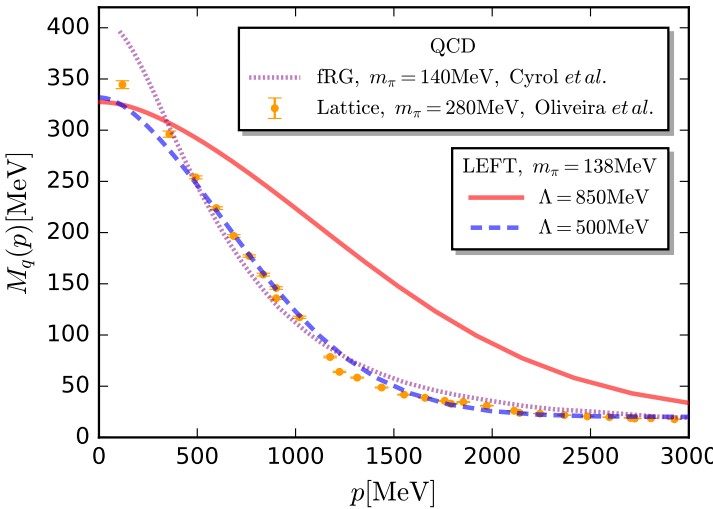

Figure 7: Light constituent quark mass $M_q(p)$ as a function of the momentum for a pion mass of $m_\pi = 138$ MeV. It is shown for two different values of the initial cutoff scale $\Lambda = 850$ MeV (red solid line) and $\Lambda = 500$ MeV (blue dashed line), in comparison to results in $N_f = 2$ flavour QCD: [13] (fRG, $m_\pi = 140$ MeV) and [37] (lattice, $m_\pi = 280$ MeV).

Note that we have re-iterated the discussion in Section 2 here, as $\lambda^-_{\sigma,\pi}(\boldsymbol{p})$ turns out to be relevant, see Figure 8. There we show the full quark mass function in comparison to that computed with only the scalar-pseudoscalar dressings $\lambda_{\sigma,\pi}(\boldsymbol{p})$. Finally, the importance of the $p^2$-tensor structures is visualised by the comparison to the quark mass function, obtained by only considering $\lambda^+(\boldsymbol{p})$ or only $\lambda^+_{\sigma,\pi}(\boldsymbol{p})$. While the difference between considering all $\lambda^+$ or only $\lambda^+_{\sigma,\pi}$ is negligible, evidently $\lambda^-_{\sigma,\pi}$ is relevant. In summary, all tensor structures but the scalar-pseudoscalar ones are negligible, as is also visible from Figure 9. In turn, the symmetric scalar-pseudoscalar momentum-squared tensor structure related to $\lambda^-_{\sigma,\pi}(\boldsymbol{p})$ is relevant. For the $t$-channel this is also illustrated in Figure 10, where the sizable contribution of $\lambda^-_\pi$ is clearly visible.

The result for the quark mass function is readily understood in terms of the ten RG-invariant dressings $\bar{\lambda}_\alpha$ of $\mathcal{T}^{(\alpha)}$, that include the full dynamics of the present approximation, see Figure 9. There we have collected all four-quark dressings (39) as functions of the momentum in the $t$-channel with $s = u = 0$. Obviously, the scalar and pseudoscalar dressings are far larger than the other dressings. While the relative size of a dressing indicates its relative importance for the dynamics of the system, a reliable statement can only be made by switching off specific channels and computing the impact of such a procedure as discussed above for the quark mass function depicted in Figure 8. Evidently, only the couplings of the $(V-A)^{\text{adj}}$, $(S+P)^{\text{adj}}_-$ and $(V-A)$ channels can play a subleading role, see the right panel of Figure 9. Figure 8 confirms that even these tensor structures are negligible. Moreover, the further channels are completely negligible. This finding is consistent with the relevant results in [1], where a momentum-independent truncation for the four-quark vertices is used.

So far we have only shown results for $t$-channel momenta with $s = u = 0$. This channel carries the dominant momentum dependence as the mesons are defined as resonant $t$-channel degrees of freedom. The most important channel is the $\pi$-channel and we now evaluate the momentum dependence of the four-quark dressings $\bar{\lambda}_\pi$ in all channels. The respective results are shown in Figures 10 to 12. The symmetry relations (A.9) and (A.11) for the $\lambda^\pm_\alpha$ imply for

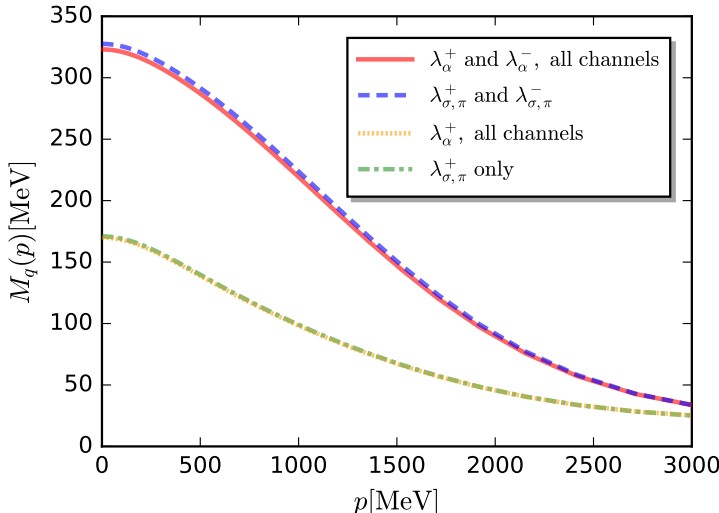

Figure 8: Comparison of the full quark mass function with that obtained only with the dressings $\lambda_{\sigma,\pi}$. We also compare that obtained with and without the antisymmetric four-quark dressings. All cases use the the same initial parameters.

the $t,s,u$-dependent four-quark dressings,

$$
\begin{aligned}
\lambda_\alpha^+(s,t,u) &= \lambda_\alpha^+(s,u,t), \\
\lambda_\alpha^-(s,t,u) &= -\lambda_\alpha^-(s,u,t).
\end{aligned}
\tag{40}
$$

Consequently, we infer $\lambda_\alpha^-(p^2,0,0) = 0$ as shown in Figure 11 and

$$
\begin{aligned}
\lambda_\alpha^+(0,p^2,0) &= \lambda_\alpha^+(0,0,p^2), \\
\lambda_\alpha^-(0,p^2,0) &= -\lambda_\alpha^-(0,0,p^2),
\end{aligned}
\tag{41}
$$

which are also confirmed by comparing Figures 10 and 12.

We close our analysis with the discussion of the Bethe-Salpeter (BS) amplitude of the pion. It is defined as

$$
h_\pi(p,\cos\theta) = \lim_{P^2 \to -m_\pi^2} \left[ \lambda_\pi(P^2,p,\cos\theta)(P^2 + m_\pi^2) \right]^{1/2},
\tag{42}
$$

with the momentum configuration

$$
\begin{aligned}
P_\mu &= \sqrt{P^2}\left(1,0,0,0\right), \\
\bar{p}_\mu &= \sqrt{p^2}\left(\cos\theta,\sin\theta,0,0\right), \\
\bar{p}'_\mu &= -\sqrt{p^2}\left(\cos\theta,\sin\theta,0,0\right),
\end{aligned}
\tag{43}
$$

instead of (24). This necessitates the evaluation of the pion channel away from $t,s,u$ momenta. Now we use, that even in the present $s,t,u$-channel approximation (21) we still can access general momentum configurations in the following simple way: we use the self-consistent solution of the quark propagator and the $s,t,u$-channel vertices in the flow diagrams and simply evaluate the later or rather the integrated flow at the momentum configuration of interest.

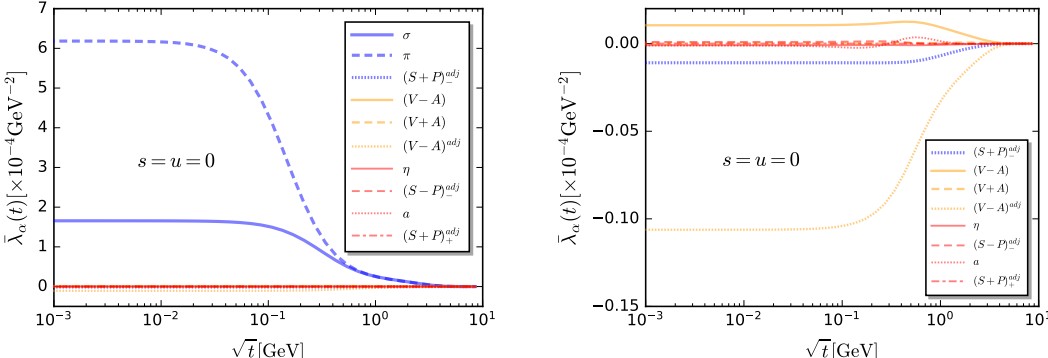

Figure 9: Left panel: RG-invariant four-quark dressings $\bar{\lambda}_\alpha = \lambda_\alpha^+ + \lambda_\alpha^-$ defined in (13), (15) and (39) of the tensors (A.1) as functions of $t$-channel momentum at $s = u = 0$. These are the combined dressings of the Fierz-complete momentum independent tensor structures and their crossing-symmetric partners, see the discussion around (39). Right panel: Zoomed-in view of the left plot without the $\sigma$ and $\pi$ dressings.

The respective numerical results are presented in Figure 13 which shows the BS amplitude as a function of $\bar{p}$ and $\cos\theta$ defined in (43). The BS amplitude only shows a very mild dependence on the angle between the quark and meson momenta, which is consistent with the results in [43]. This in turn also corroborates the $s, tu$-channel approximation for the four-quark vertices used in this work.

## 4 Conclusions and outlook

In this work we have further developed the bound state approach within the functional renormalisation group initiated in [1]. In comparison to that work we have computed the four quark dressings $\lambda_\alpha(s, t, u)$ in a $s, t, u$-channel approximation instead only in the $t$-channel. This qualitative improvement is visible in all results, and in particular it leads to a qualitatively improved convergence in the chiral limit. Deep in the chiral limit the current approximation still fails, though it fails for considerably smaller pion masses in comparison to [1]. This problem relates to the non-trivial momentum transfer in the loops inherent to the current setup. This intricacy arises from the emergence of the soft pion mode [44] below the chiral symmetry breaking scale and is easily remedied with the inclusion of emergent composites as done in [10, 11, 13, 15]. The purpose of the present work and that of [1] was complementary, concentrating on the quantitative resolution of the momentum dependence of the Fierz-complete four-quark vertex. We have also investigated the quantitative reliability of the $s, t, u$-channel approximation for the four-quark vertices. Notably, our results are in excellent agreement with the $\chi$PT at the order of $\mathcal{O}(p^4)$ in the regime of $m_\pi \lesssim 170$ MeV.

We have also discussed in detail the determination of the absolute physics scale with the ratios of the pion decay constants in the chiral limit and at the physical point with the physical pion mass. Moreover, the light constituent quark mass function has been compared with functional QCD and lattice QCD results. We find, that the momentum dependence does not match that in QCD for an ultraviolet cutoff scale $\lambda = 850$ MeV. For a UV cutoff of $\lambda = 500$ MeV, the mass function agrees reasonably well with the QCD mass function, see Figure 7. These findings indicate that the glue dynamics is still relevant in the momentum regime between 1 GeV and 500 MeV, which corroborated similar findings in [14, 42]. The respective analysis will be furthered in [27] within functional QCD.

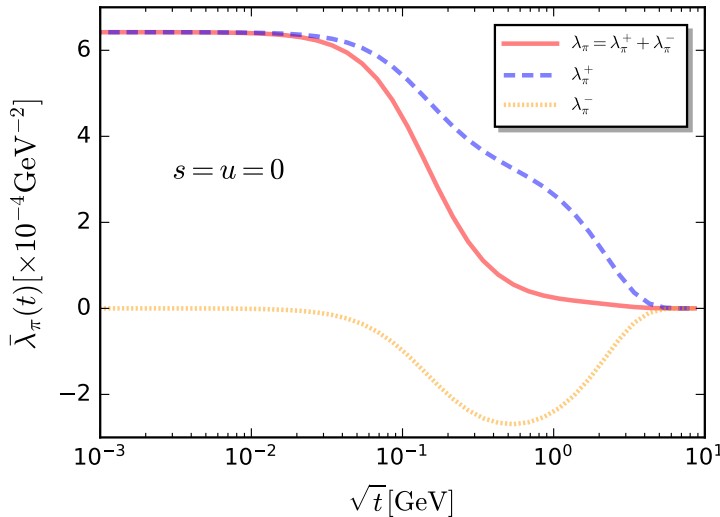

Figure 10: Symmetric, antisymmetric and total four-quark couplings in the $\pi$ channel, i.e., $\lambda_\pi^+$, $\lambda_\pi^-$, $\lambda_\pi$, respectively, as functions of the $t$-channel momentum at $s = u = 0$. The $s, u$-channel results are depicted in Figures 11 and 12.

The current work paves the way for the final step towards quantitative predictions of the QCD resonance structure and general timelike correlation functions in first-principles functional QCD with the fRG. In this step we will add the pure glue sector and the quark-gluon vertex as the interface between the matter sector discussed here and the pure glue sector, for quantitative vacuum results see [45–47]. A final improvement of the momentum dependence is provided by the inclusion of a symmetric point part of the momentum structure: While the $s, t, u$-channel approximation is well adapted to resonances, the symmetric point approximation deals well with resonance-free dressings. In summary, the present advances in the four-quark sector of QCD, together with the fRG with emergent composites allow for a full systematic resolution of the infrared dynamics of QCD including the resonance structure and we hope to report on respective results in the near future.

## Acknowledgments

We thank Zhi-Hui Guo for discussions.

**Funding information** This work is supported by the National Natural Science Foundation of China under Grant Nos. 12175030, 12147101. This work is also funded by the Deutsche Forschungsgemeinschaft (DFG, German Research Foundation) under Germany's Excellence Strategy EXC 2181/1 - 390900948 (the Heidelberg STRUCTURES Excellence Cluster) and the Collaborative Research Centre SFB 1225 - 273811115 (ISOQUANT).

## A Fierz-complete basis and symmetry relations of the four-quark vertex

In this appendix we summarise the tensor structures of the Fierz-complete basis we use in the current work as well as discuss symmetry relations. This basis, within minor modifications, has also been used in our previous work [1] and in related fRG works [11,13,32], see also the review [28].

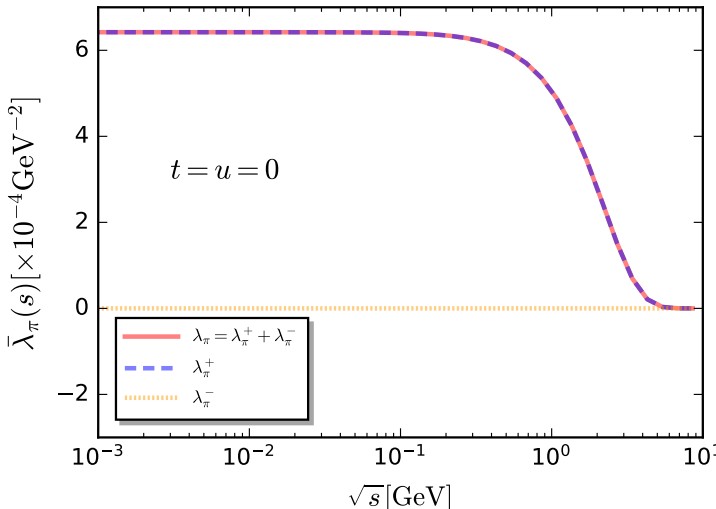

Figure 11: Symmetric, antisymmetric and total four-quark couplings in the $\pi$ channel, i.e., $\lambda_\pi^+$, $\lambda_\pi^-$, $\lambda_\pi$, respectively, as functions of the $s$-channel momentum with $t = u = 0$. The $t, u$-channel results are depicted in Figures 10 and 12.

The Fierz-complete basis of momentum-independent tensor structures is built from the ten tensors

$$
\begin{aligned}
\mathcal{T}^\sigma_{ijlm} &= (T^0)_{ij}(T^0)_{lm}, \\
\mathcal{T}^\pi_{ijlm} &= -(\gamma_5 T^a)_{ij}(\gamma_5 T^a)_{lm}, \\
\mathcal{T}^a_{ijlm} &= (T^a)_{ij}(T^a)_{lm}, \\
\mathcal{T}^\eta_{ijlm} &= -(\gamma_5 T^0)_{ij}(\gamma_5 T^0)_{lm}, \\
\mathcal{T}^{(V-A)}_{ijlm} &= (\gamma_\mu T^0)_{ij}(\gamma_\mu T^0)_{lm} - (i\gamma_\mu\gamma_5 T^0)_{ij}(i\gamma_\mu\gamma_5 T^0)_{lm}, \\
\mathcal{T}^{(V+A)}_{ijlm} &= (\gamma_\mu T^0)_{ij}(\gamma_\mu T^0)_{lm} + (i\gamma_\mu\gamma_5 T^0)_{ij}(i\gamma_\mu\gamma_5 T^0)_{lm}, \\
\mathcal{T}^{(V-A)^{\mathrm{adj}}}_{ijlm} &= (\gamma_\mu T^0 t^a)_{ij}(\gamma_\mu T^0 t^a)_{lm} - (i\gamma_\mu\gamma_5 T^0 t^a)_{ij}(i\gamma_\mu\gamma_5 T^0 t^a)_{lm}, \\
\mathcal{T}^{(S+P)^{\mathrm{adj}}_-}_{ijlm} &= (T^0 t^a)_{ij}(T^0 t^a)_{lm} + (\gamma_5 T^0 t^a)_{ij}(\gamma_5 T^0 t^a)_{lm} \\
&\quad - (T^a t^b)_{ij}(T^a t^b)_{lm} - (\gamma_5 T^a t^b)_{ij}(\gamma_5 T^a t^b)_{lm}, \\
\mathcal{T}^{(S-P)^{\mathrm{adj}}_-}_{ijlm} &= (T^0 t^a)_{ij}(T^0 t^a)_{lm} - (\gamma_5 T^0 t^a)_{ij}(\gamma_5 T^0 t^a)_{lm} \\
&\quad - (T^a t^b)_{ij}(T^a t^b)_{lm} + (\gamma_5 T^a t^b)_{ij}(\gamma_5 T^a t^b)_{lm}, \\
\mathcal{T}^{(S+P)^{\mathrm{adj}}_+}_{ijlm} &= (T^0 t^a)_{ij}(T^0 t^a)_{lm} + (\gamma_5 T^0 t^a)_{ij}(\gamma_5 T^0 t^a)_{lm} \\
&\quad + (T^a t^b)_{ij}(T^a t^b)_{lm} + (\gamma_5 T^a t^b)_{ij}(\gamma_5 T^a t^b)_{lm}.
\end{aligned}
\tag{A.1}
$$

The part of the full four-quark term derived from these tensor structures reads,

$$
\Gamma_{4q,k} = -\int \frac{d^4 p_1}{(2\pi)^4} \cdots \frac{d^4 p_4}{(2\pi)^4}(2\pi)^4 \delta^4(\boldsymbol{p}) \sum_{\alpha\in\mathcal{F}} \lambda_\alpha(\boldsymbol{p})\, \mathcal{T}^{(\alpha)}_{ijlm}(\boldsymbol{p})\, \bar{q}_i(p_1) q_j(p_2) \bar{q}_l(p_3) q_m(p_4),
\tag{A.2}
$$

with

$$
\mathcal{F} = \left\{ \sigma, \pi, a, \eta, (V\pm A), (V-A)^{\mathrm{adj}}, (S\pm P)^{\mathrm{adj}}_-, (S+P)^{\mathrm{adj}}_+ \right\},
\tag{A.3}
$$

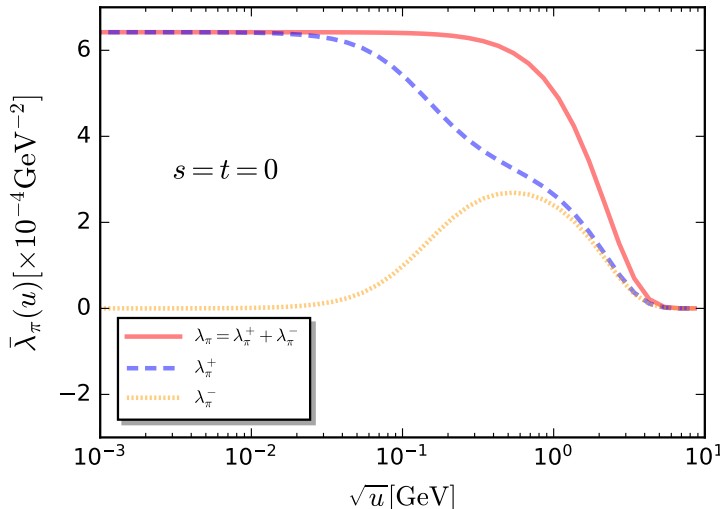

Figure 12: Symmetric, antisymmetric and total four-quark couplings in the $\pi$ channel, i.e., $\lambda_\pi^+$, $\lambda_\pi^-$, $\lambda_\pi$, respectively, as functions of the $u$-channel momentum, $\sqrt{u}$ at $s = t = 0$. The $s, t$-channel results are depicted in Figures 10 and 11.

and the momentum-dependent dressings $\lambda_\alpha(\boldsymbol{p})$ with $\boldsymbol{p} = (p_1, ..., p_4)$, see (8). Crossing symmetry (13) entails

$$\mathcal{T}_{ijlm}^{(\alpha)} = \mathcal{T}_{lmij}^{(\alpha)},$$
$$\lambda_\alpha(p_1, p_2, p_3, p_4) = \lambda_\alpha(p_3, p_4, p_1, p_2). \tag{A.4}$$

The ten tensors $\mathcal{T}^\alpha$ in (A.1) can be split into their symmetric and antisymmetric components,

$$\mathcal{T}_{ijlm}^{(\alpha)} = \mathcal{T}_{ijlm}^{(\alpha^+)} + \mathcal{T}_{ijlm}^{(\alpha^-)},$$
$$\mathcal{T}_{ijlm}^{(\alpha^\pm)} = \frac{1}{2}\Big(\mathcal{T}_{ijlm}^{(\alpha)} \pm \mathcal{T}_{ljim}^{(\alpha)}\Big), \tag{A.5}$$

with

$$\mathcal{T}_{ijlm}^{(\alpha^+)} = \mathcal{T}_{ljim}^{(\alpha^+)} = \mathcal{T}_{imlj}^{(\alpha^+)} = \mathcal{T}_{lmij}^{(\alpha^+)},$$
$$\mathcal{T}_{ijlm}^{(\alpha^-)} = -\mathcal{T}_{ljim}^{(\alpha^-)} = -\mathcal{T}_{imlj}^{(\alpha^-)} = \mathcal{T}_{lmij}^{(\alpha^-)}. \tag{A.6}$$

In (A.2) they come with the respective dressings $\lambda^{(\alpha^\pm)}$,

$$\lambda_\alpha^+(p_1, p_2, p_3, p_4) \equiv \frac{1}{2}\Big[\lambda_\alpha(p_1, p_2, p_3, p_4) + \lambda_\alpha(p_3, p_2, p_1, p_4)\Big], \tag{A.7}$$

$$\lambda_\alpha^-(p_1, p_2, p_3, p_4) \equiv \frac{1}{2}\Big[\lambda_\alpha(p_1, p_2, p_3, p_4) - \lambda_\alpha(p_3, p_2, p_1, p_4)\Big], \tag{A.8}$$

whose symmetry under momentum permutations, (A.9) and (A.11), follows from that of the respective tensors.

In particular, upon contraction with the quarks and antiquarks only the antisymmetric components $\mathcal{T}^{(\alpha^-)}$ of the above tensors survive for vanishing momenta, and $\{\mathcal{T}^{(\alpha^-)}\}$ constitutes a Fierz-complete basis of the momentum-independent tensor structures. Due to the antisymmetry of the $\mathcal{T}^{(\alpha^-)}$, their dressings $\lambda_\alpha^+$ of the tensors are positive under commutation of momenta,

$$\lambda_\alpha^+(p_1, p_2, p_3, p_4) = \lambda_\alpha^+(p_3, p_2, p_1, p_4) = \lambda_\alpha^+(p_1, p_4, p_3, p_2) = \lambda_\alpha^+(p_3, p_4, p_1, p_2), \tag{A.9}$$

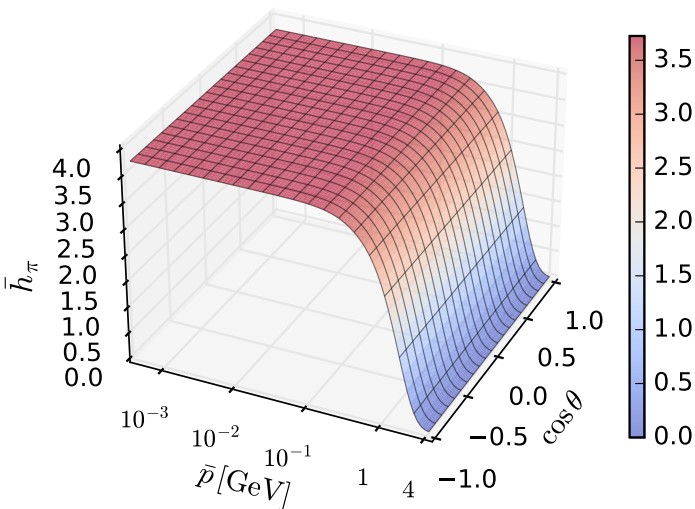

Figure 13: Bethe-Salpeter amplitude of the pion as a function of the magnitude of the quark momentum and the angle between the quark and meson momenta.

which completes the construction of the four-quark scattering term with momentum-independent tensor structures. The respective four-quark term in the effective action is obtained by simply substituting

$$\mathcal{T}^{(\alpha)}\lambda^{(\alpha)} \to \mathcal{T}^{(\alpha^-)}\lambda^{(\alpha^+)}, \tag{A.10}$$

in (A.2). A natural extension of the above Fierz-complete basis of momentum independent tensor structures is given by the symmetric parts of $\mathcal{T}$ with the dressings

$$\lambda_\alpha^-(p_1, p_2, p_3, p_4) = -\lambda_\alpha^-(p_3, p_2, p_1, p_4) = -\lambda_\alpha^-(p_1, p_4, p_3, p_2) = \lambda_\alpha^-(p_3, p_4, p_1, p_2), \tag{A.11}$$

that vanish at $\boldsymbol{p} = 0$. Accordingly, the part $\mathcal{T}^{(\alpha^+)}\lambda^{(\alpha^-)}$ comprises a part of the four-quark term built from momentum-dependent tensor structures. It is required for maintaining the crossing symmetry of (A.2) for $\boldsymbol{p} \neq 0$. We emphasise that it is neither complete nor unique and depends on the choice of the basis tensors in (A.1), while the momentum-independent part is complete.

# B    Flows of the four-quark vertices

The flow equations for the quark self-energy and the four-quark vertices are shown in Figure 14. First of all, we focus on the flows of four-quark vertices. The flows of the symmetric and antisymmetric four-quark dressings in (A.7) and (A.8) can be obtained by projecting the flows of vertices onto the tensors (10) and (11), respectively. Since Fierz-complete basis of tensor structures is employed, there is no ambiguity for the projections, see [1] for more details. The flows of four-quark dressings can be decomposed into a sum of $s, t, u$-channel diagrams,

$$\partial_t \lambda_\alpha^\pm(p_1, p_2, p_3, p_4) = \sum_{i=s,t,u} \text{Flow}_{(\lambda_\alpha^\pm)}^{(i)}(p_1, p_2, p_3, p_4), \tag{B.1}$$

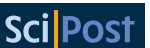

$$\partial_t \left( \rightarrow\!\!\!\bullet\!\!\!\rightarrow \right) = \tilde{\partial}_t \left( -\rightarrow\!\!\bigcirc\!\!\rightarrow \right)$$

$$\partial_t \left( \!\!\times\!\! \right) = \tilde{\partial}_t \left( -\!\!\!\! \rtimes\!\!\ltimes \!+ \rtimes\!\!\ltimes + \frac{1}{2}\!\!\times \right)$$

Figure 14: Flow equations for the two- and four-point quark correlation functions, see also [1]. The blobs stand for 1PI $n$-point functions, see Figure 15, the derivative $\tilde{\partial}_t$ only hits the $k$-dependence of the regulators, while the tilde-derivative of the $n$-point functions vanishes, $\tilde{\partial}_t \Gamma^{(n)} \equiv 0$. The inner lines depict full momentum-dependent propagators.

$$-\Gamma^{(n)}_{\Phi_{i_1}\cdots\Phi_{i_n}}(p_1,\cdots,p_n) =$$ 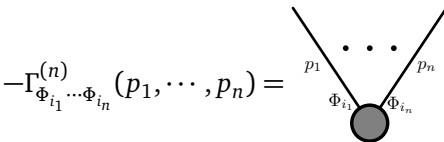

Figure 15: Diagrammatic representation of general 1PI $n$-point functions or vertices. All momenta are counted incoming and the fields $\Phi_{i_j}$ label the field of the respective leg.

see also Figures 14 and 16. The flow of $t$-channel reads

$$\text{Flow}^{(t)}_{(\lambda_a)}\Big(-\bar{p}-P/2, \bar{p}-P/2, -\bar{p}'+P/2, \bar{p}'+P/2\Big)$$
$$= \sum_{a',a''} \int \frac{d^4q}{(2\pi)^4} \lambda_{a'}\Big(-\bar{p}-P/2, \bar{p}-P/2, -q, q+P\Big)$$
$$\times \lambda_{a''}\Big(-q-P, q, -\bar{p}'+P/2, \bar{p}'+P/2\Big)\mathcal{F}^t_{a'a'',a}$$
$$= \sum_{a',a''} \int \frac{d^4q}{(2\pi)^4} \lambda_{a'}(t', u', s')\lambda_{a''}(t'', u'', s'')\mathcal{F}^t_{a'a'',a}, \tag{B.2}$$

with

$$s' = (q+\bar{p}+P/2)^2, \qquad t' = P^2, \qquad u' = (q-\bar{p}+P/2)^2, \tag{B.3}$$

and

$$s'' = \big(q+\bar{p}'+P/2\big)^2, \qquad t'' = P^2, \qquad u'' = \big(q-\bar{p}'+P/2\big)^2, \tag{B.4}$$

where we have relabelled the notation $\{\lambda_a\} = \{\lambda^\pm_\alpha\}$ and used the three-momentum-channel approximation for the four-quark vertices in (21) in the last line of (B.2). The coefficients $\mathcal{F}^t_{a'a'',a}$ as well as $\mathcal{F}^u_{a'a'',a}$ and $\mathcal{F}^s_{a'a'',a}$ in what follows are momentum-dependent functions of the quark propagators and regulators. The momentum routing for the $t$-channel flow diagram of the four-quark vertex is shown in Figure 16, together with those for the other $u$ and $s$ channels. The loop momentum in (B.2) reads

$$q_\mu = \sqrt{q^2}\Big(\cos\theta_1, \sin\theta_1\cos\theta_2, \sin\theta_1\sin\theta_2\cos\varphi, \sin\theta_1\sin\theta_2\sin\varphi\Big), \tag{B.5}$$

where $\theta_1, \theta_2 \in [0,\pi]$ and $\varphi \in [0,2\pi]$. The integral measure is given by

$$\int d^4q = \int_0^\infty dq\, q^3 \int_0^\pi d\theta_1 \sin^2\theta_1 \int_0^\pi d\theta_2 \sin\theta_2 \int_0^{2\pi} d\varphi. \tag{B.6}$$

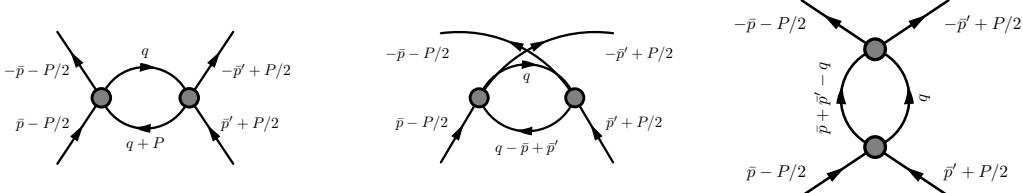

Figure 16: Momentum routing of the flow diagrams of four-quark vertices for the $t$-, $u$-, and $s$-momentum channels, respectively.

For the sake of brevity, we adopt the shorthand notations

$$
\begin{aligned}
\boldsymbol{t}_1 &\equiv \left((q+\bar{p}+P/2)^2, P^2, (q-\bar{p}+P/2)^2\right), \\
\boldsymbol{t}_2 &\equiv \left((q+\bar{p}'+P/2)^2, P^2, (q-\bar{p}'+P/2)^2\right).
\end{aligned}
\tag{B.7}
$$

In the same way, for the flow diagrams of the $u$ and $s$ channels in Figure 16, we define

$$
\begin{aligned}
\boldsymbol{u}_1 &\equiv \left((q+\bar{p}'-P/2)^2, (\bar{p}-\bar{p}')^2, (q-\bar{p}+P/2)^2\right), \\
\boldsymbol{u}_2 &\equiv \left((q+\bar{p}'+P/2)^2, (\bar{p}-\bar{p}')^2, (q-\bar{p}-P/2)^2\right),
\end{aligned}
\tag{B.8}
$$

and

$$
\begin{aligned}
\boldsymbol{s}_1 &\equiv \left((\bar{p}+\bar{p}')^2, (q-\bar{p}'+P/2)^2, (q-\bar{p}-P/2)^2\right), \\
\boldsymbol{s}_2 &\equiv \left((\bar{p}+\bar{p}')^2, (q-\bar{p}'-P/2)^2, (q-\bar{p}+P/2)^2\right).
\end{aligned}
\tag{B.9}
$$

Consequently, the flow of four-quark dressings for the $u$-channel in (B.1) reads

$$
\mathrm{Flow}^{(u)}_{(\lambda_a)}\left(-\bar{p}-P/2, \bar{p}-P/2, -\bar{p}'+P/2, \bar{p}'+P/2\right) = \sum_{a',a''}\int \frac{d^4q}{(2\pi)^4}\lambda_{a'}(\boldsymbol{u}_1)\lambda_{a''}(\boldsymbol{u}_2)\mathcal{F}^u_{a'a'',a}, \tag{B.10}
$$

and that of the $s$-channel reads

$$
\mathrm{Flow}^{(s)}_{(\lambda_a)}\left(-\bar{p}-P/2, \bar{p}-P/2, -\bar{p}'+P/2, \bar{p}'+P/2\right) = \sum_{a',a''}\int \frac{d^4q}{(2\pi)^4}\lambda_{a'}(\boldsymbol{s}_1)\lambda_{a''}(\boldsymbol{s}_2)\mathcal{F}^s_{a'a'',a}. \tag{B.11}
$$

The explicit expressions of the four-quark flows in (B.1) are very lengthy and we refrain from presenting them here. Instead, we provide the flows that follow from only including the $\sigma$ and $\pi$ tensor structures. It has been discussed in Section 3.2, that these flows already provide quantitatively reliable results.

The flow of $t$-channel for the symmetric dressing of the $\sigma$ tensor structure is given by

$$
\begin{aligned}
&\mathrm{Flow}^{(t)}_{(\lambda_\sigma^+)}\left(-\bar{p}-P/2, \bar{p}-P/2, -\bar{p}'+P/2, \bar{p}'+P/2\right) \\
&= \int \frac{d^4q}{(2\pi)^4}\frac{\bar{t}_s(q)}{Z_q(q)Z_q(q_t)}\bar{G}_s(q_t)\left\{q^2\left[1+r_q\left(\frac{q^2}{k^2}\right)\right]M_q(q)M_q(q_t)\mathcal{A}^{(t)}_{(\lambda_\sigma^+)}(\boldsymbol{t}_1, \boldsymbol{t}_2)\right. \\
&\left.\qquad + \frac{q\cdot q_t}{24}\left[1+r_q\left(\frac{q_t^2}{k^2}\right)\right]\left[M_q^2(q)-q^2\left[1+r_q\left(\frac{q^2}{k^2}\right)\right]^2\right]\mathcal{B}^{(t)}_{(\lambda_\sigma^+)}(\boldsymbol{t}_1, \boldsymbol{t}_2)\right\},
\end{aligned}
\tag{B.12}
$$

with $q_t = q+P$ denoting the loop momentum of one of the inner quark propagators as shown in the first diagram of Figure 16. Here the scalar part of the RG invariant propagator $\bar{G}_s$ is given by

$$
\bar{G}_s(q) = \frac{1}{q^2\left[1+r_q\left(\frac{q^2}{k^2}\right)\right]^2 + M_q^2(q)}, \tag{B.13}
$$

and the RG-invariant part of the single-scale propagator reads

$$\bar{t}_s(q) = \bar{G}_s(q) \left[ \left( \partial_t - \eta_q(q) \right) r_q \left( \frac{q^2}{k^2} \right) \right] \bar{G}_s(q), \tag{B.14}$$

with the shape function $r_q(q^2/k^2)$ of the regulator $R_q(q)$ defined in (4) and the anomalous dimension $\eta_q$,

$$\eta_q(q) = -\frac{\partial_t Z_q(q)}{Z_q(q)}. \tag{B.15}$$

The functions $\mathcal{A}$ and $\mathcal{B}$ in (B.12) are given by

$$
\begin{aligned}
\mathcal{A}^{(t)}_{(\lambda_\sigma^+)}(t_1, t_2) &= 25\lambda_\sigma^-(t_1)\lambda_\sigma^-(t_2) + 3\lambda_\pi^+(t_1)\lambda_\sigma^-(t_2) + 3\lambda_\sigma^-(t_1)\lambda_\pi^+(t_2) - 3\lambda_\pi^+(t_1)\lambda_\pi^+(t_2) \\
&\quad + 25\lambda_\pi^+(t_1)\lambda_\sigma^-(t_2) + 3\lambda_\sigma^+(t_1)\lambda_\pi^+(t_2) - 3\lambda_\sigma^-(t_1)\lambda_\pi^-(t_2) + 3\lambda_\pi^+(t_1)\lambda_\pi^-(t_2) \\
&\quad - 3\lambda_\sigma^+(t_1)\lambda_\pi^-(t_2) + 25\lambda_\sigma^-(t_1)\lambda_\sigma^+(t_2) + 3\lambda_\pi^+(t_1)\lambda_\sigma^+(t_2) + 21\lambda_\sigma^+(t_1)\lambda_\sigma^+(t_2) \\
&\quad - 3\lambda_\pi^-(t_1)\lambda_\pi^-(t_2) - 3\lambda_\pi^-(t_1)\lambda_\sigma^-(t_2) + 3\lambda_\pi^-(t_1)\lambda_\pi^+(t_2) - 3\lambda_\pi^-(t_1)\lambda_\sigma^+(t_2), \tag{B.16}
\end{aligned}
$$

and

$$
\begin{aligned}
\mathcal{B}^{(t)}_{(\lambda_\sigma^+)}(t_1, t_2) &= 312\lambda_\sigma^-(t_1)\lambda_\sigma^-(t_2) + 37\lambda_\pi^+(t_1)\lambda_\sigma^-(t_2) + 37\lambda_\sigma^-(t_1)\lambda_\pi^+(t_2) + 2\lambda_\pi^+(t_1)\lambda_\pi^+(t_2) \\
&\quad + 288\lambda_\sigma^+(t_1)\lambda_\sigma^-(t_2) + 35\lambda_\sigma^+(t_1)\lambda_\pi^+(t_2) - 37\lambda_\sigma^-(t_1)\lambda_\pi^-(t_2) - 2\lambda_\pi^+(t_1)\lambda_\pi^-(t_2) \\
&\quad - 35\lambda_\sigma^+(t_1)\lambda_\pi^-(t_2) + 2\lambda_\pi^-(t_1)\lambda_\pi^-(t_2) - 37\lambda_\pi^-(t_1)\lambda_\sigma^-(t_2) - 2\lambda_\pi^-(t_1)\lambda_\pi^+(t_2) \\
&\quad - 35\lambda_\pi^-(t_1)\lambda_\sigma^+(t_2) + 288\lambda_\sigma^-(t_1)\lambda_\sigma^+(t_2) + 35\lambda_\pi^+(t_1)\lambda_\sigma^+(t_2) + 264\lambda_\sigma^+(t_1)\lambda_\sigma^+(t_2). \tag{B.17}
\end{aligned}
$$

The flow of $u$-channel can be directly obtained from that of $t$ one with the interchange of the momenta of two quarks or two antiquarks. Thus, one is led to

$$\text{Flow}^{(u)}_{(\lambda_\sigma^+)}\left( -\bar{p} - P/2, \bar{p} - P/2, -\bar{p}' + P/2, \bar{p}' + P/2 \right) = \text{Flow}^{(t)}_{(\lambda_\sigma^+)} \Big|_{q_t \to q_u, t_1 \to u_1, t_2 \to u_2}, \tag{B.18}$$

with $q_u = q - \bar{p} + \bar{p}'$. The flow of $s$-channel for the $\lambda_\sigma^+$ reads

$$
\begin{aligned}
&\text{Flow}^{(s)}_{(\lambda_\sigma^+)}\left( -\bar{p} - P/2, \bar{p} - P/2, -\bar{p}' + P/2, \bar{p}' + P/2 \right) \\
&= \int \frac{d^4 q}{(2\pi)^4} \frac{\bar{t}_s(q)}{Z_q(q)Z_q(q_s)} \bar{G}_s(q_s) \left\{ -2q^2 \left[ 1 + r_q\left(\frac{q^2}{k^2}\right) \right] M_q(q) M_q(q_s) \left( 3\lambda_\pi^+(s_1)\lambda_\pi^+(s_2) + \lambda_\sigma^+(s_1)\lambda_\sigma^+(s_2) \right) \right. \tag{B.19} \\
&\quad \left. + \frac{q \cdot q_s}{12} \left[ 1 + r_q\left(\frac{q_s^2}{k^2}\right) \right] \left[ M_q^2(q) - q^2 \left[ 1 + r_q\left(\frac{q^2}{k^2}\right) \right] \right]^2 \left( 2\lambda_\pi^+(s_1)\lambda_\pi^+(s_2) - \lambda_\sigma^+(s_1)\lambda_\pi^+(s_2) - \lambda_\pi^+(s_1)\lambda_\sigma^+(s_2) \right) \right\},
\end{aligned}
$$

with $q_s = \bar{p} + \bar{p}' - q$. The flow of $t$-channel for $\lambda_\pi^+$ is given by

$$
\begin{aligned}
&\text{Flow}^{(t)}_{(\lambda_\pi^+)}\left( -\bar{p} - P/2, \bar{p} - P/2, -\bar{p}' + P/2, \bar{p}' + P/2 \right) \\
&= \int \frac{d^4 q}{(2\pi)^4} \frac{\bar{t}_s(q)}{Z_q(q)Z_q(q_t)} \bar{G}_s(q_t) \left\{ -2q^2 \left[ 1 + r_q\left(\frac{q^2}{k^2}\right) \right] M_q(q) M_q(q_t) \mathcal{A}^{(t)}_{(\lambda_\pi^+)}(t_1, t_2) \right. \tag{B.20} \\
&\quad \left. + \frac{q \cdot q_t}{24} \left[ 1 + r_q\left(\frac{q_t^2}{k^2}\right) \right] \left[ M_q^2(q) - q^2 \left[ 1 + r_q\left(\frac{q^2}{k^2}\right) \right] \right]^2 \mathcal{B}^{(t)}_{(\lambda_\pi^+)}(t_1, t_2) \right\},
\end{aligned}
$$

with

$$
\begin{aligned}
\mathcal{A}^{(t)}_{(\lambda_\pi^+)}(t_1, t_2) &= 11\lambda_\pi^-(t_1)\lambda_\pi^-(t_2) + 12\lambda_\pi^+(t_1)\lambda_\pi^-(t_2) - \lambda_\pi^+(t_1)\lambda_\sigma^-(t_2) + 12\lambda_\pi^-(t_1)\lambda_\pi^+(t_2) \\
&\quad - \lambda_\sigma^-(t_1)\lambda_\pi^+(t_2) + 13\lambda_\pi^+(t_1)\lambda_\pi^+(t_2) + \lambda_\sigma^+(t_1)\lambda_\pi^+(t_2) + \lambda_\pi^+(t_1)\lambda_\sigma^+(t_2), \tag{B.21}
\end{aligned}
$$

and

$$\mathcal{B}^{(t)}_{(\lambda^+_\pi)}(t_1, t_2) = 310\lambda^+_\pi(t_1)\lambda^+_\pi(t_2) - 13\lambda^+_\pi(t_1)\lambda^-_\sigma(t_2) - 13\lambda^-_\sigma(t_1)\lambda^+_\pi(t_2) + 13\lambda^+_\sigma(t_1)\lambda^+_\pi(t_2)$$

$$- 11\lambda^-_\sigma(t_1)\lambda^-_\pi(t_2) + 290\lambda^+_\pi(t_1)\lambda^-_\pi(t_2) + 11\lambda^+_\sigma(t_1)\lambda^-_\pi(t_2) + 13\lambda^+_\pi(t_1)\lambda^+_\sigma(t_2)$$

$$+ 262\lambda^-_\pi(t_1)\lambda^-_\pi(t_2) - 11\lambda^-_\pi(t_1)\lambda^-_\sigma(t_2) + 290\lambda^-_\pi(t_1)\lambda^+_\pi(t_2) + 11\lambda^-_\pi(t_1)\lambda^+_\sigma(t_2), \quad \text{(B.22)}$$

Similar with (B.18), one has

$$\text{Flow}^{(u)}_{(\lambda^+_\pi)}\left(-\bar{p} - P/2, \bar{p} - P/2, -\bar{p}' + P/2, \bar{p}' + P/2\right) = \text{Flow}^{(t)}_{(\lambda^+_\pi)}\Big|_{q_t \to q_u, t_1 \to u_1, t_2 \to u_2}. \quad \text{(B.23)}$$

The flow of $s$-channel for $\lambda^+_\pi$ reads

$$\text{Flow}^{(s)}_{(\lambda^+_\pi)}\left(-\bar{p} - P/2, \bar{p} - P/2, -\bar{p}' + P/2, \bar{p}' + P/2\right)$$

$$= \int \frac{d^4q}{(2\pi)^4} \frac{\bar{t}_s(q)}{Z_q(q)Z_q(q_s)} \bar{G}_s(q_s) \left\{ -2q^2 \left[1 + r_q\left(\frac{q^2}{k^2}\right)\right] M_q(q)M_q(q_s)\left(\lambda^+_\pi(s_1)\lambda^+_\pi(s_2) + \lambda^+_\sigma(s_1)\lambda^+_\sigma(s_2)\right) \right. \quad \text{(B.24)}$$

$$\left. + \frac{q \cdot q_s}{12} \left[1 + r_q\left(\frac{q^2_s}{k^2}\right)\right]\left[M^2_q(q) - q^2\left[1 + r_q\left(\frac{q^2}{k^2}\right)\right]\right]^2 \left(\lambda^+_\sigma(s_1)\lambda^+_\pi(s_2) + \lambda^+_\pi(s_1)\lambda^+_\sigma(s_2) - 2\lambda^+_\pi(s_1)\lambda^+_\pi(s_2)\right) \right\}.$$

It is left to present the flows of the antisymmetric four-quark dressings. We begin with the flow of $t$-channel for the $\lambda^-_\sigma$, that reads

$$\text{Flow}^{(t)}_{(\lambda^-_\sigma)}\left(-\bar{p} - P/2, \bar{p} - P/2, -\bar{p}' + P/2, \bar{p}' + P/2\right)$$

$$= \int \frac{d^4q}{(2\pi)^4} \frac{\bar{t}_s(q)}{Z_q(q)Z_q(q_t)} \bar{G}_s(q_t) \left\{ q^2\left[1 + r_q\left(\frac{q^2_t}{k^2}\right)\right] M_q(q)M_q(q_t)\mathcal{A}^{(t)}_{(\lambda^-_\sigma)}(t_1, t_2) \right.$$

$$\left. + \frac{q \cdot q_t}{24} \left[1 + r_q\left(\frac{q^2_t}{k^2}\right)\right]\left[M^2_q(q) - q^2\left[1 + r_q\left(\frac{q^2}{k^2}\right)\right]\right]^2 \mathcal{B}^{(t)}_{(\lambda^-_\sigma)}(t_1, t_2) \right\}, \quad \text{(B.25)}$$

with

$$\mathcal{A}^{(t)}_{(\lambda^-_\sigma)}(t_1, t_2) = 27\lambda^-_\sigma(t_1)\lambda^-_\sigma(t_2) + 3\lambda^+_\pi(t_1)\lambda^-_\sigma(t_2) + 3\lambda^-_\sigma(t_1)\lambda^+_\pi(t_2) + 3\lambda^+_\pi(t_1)\lambda^+_\pi(t_2)$$

$$+ 23\lambda^+_\pi(t_1)\lambda^-_\sigma(t_2) + 3\lambda^+_\sigma(t_1)\lambda^-_\pi(t_2) - 3\lambda^-_\sigma(t_1)\lambda^-_\pi(t_2) - 3\lambda^+_\pi(t_1)\lambda^-_\pi(t_2)$$

$$- 3\lambda^+_\sigma(t_1)\lambda^-_\pi(t_2) + 23\lambda^-_\sigma(t_1)\lambda^+_\pi(t_2) + 3\lambda^+_\pi(t_1)\lambda^+_\sigma(t_2) + 23\lambda^+_\sigma(t_1)\lambda^+_\sigma(t_2)$$

$$+ 3\lambda^-_\pi(t_1)\lambda^-_\pi(t_2) - 3\lambda^-_\pi(t_1)\lambda^-_\sigma(t_2) - 3\lambda^-_\pi(t_1)\lambda^+_\pi(t_2) - 3\lambda^-_\pi(t_1)\lambda^+_\sigma(t_2), \quad \text{(B.26)}$$

and

$$\mathcal{B}^{(t)}_{(\lambda^-_\sigma)}(t_1, t_2) = 312\lambda^-_\sigma(t_1)\lambda^-_\sigma(t_2) + 37\lambda^+_\pi(t_1)\lambda^-_\sigma(t_2) + 37\lambda^-_\sigma(t_1)\lambda^+_\pi(t_2) + 2\lambda^+_\pi(t_1)\lambda^+_\pi(t_2)$$

$$+ 288\lambda^+_\sigma(t_1)\lambda^-_\sigma(t_2) + 35\lambda^+_\sigma(t_1)\lambda^+_\pi(t_2) - 37\lambda^-_\sigma(t_1)\lambda^-_\pi(t_2) - 2\lambda^+_\pi(t_1)\lambda^-_\pi(t_2)$$

$$- 35\lambda^+_\sigma(t_1)\lambda^-_\pi(t_2) + 2\lambda^-_\pi(t_1)\lambda^-_\pi(t_2) - 37\lambda^-_\pi(t_1)\lambda^-_\sigma(t_2) - 2\lambda^-_\pi(t_1)\lambda^+_\pi(t_2)$$

$$- 35\lambda^-_\pi(t_1)\lambda^+_\sigma(t_2) + 288\lambda^-_\sigma(t_1)\lambda^-_\sigma(t_2) + 35\lambda^+_\pi(t_1)\lambda^+_\sigma(t_2)$$

$$+ 264\lambda^+_\sigma(t_1)\lambda^+_\sigma(t_2). \quad \text{(B.27)}$$

This result also allows us to obtain the flow of $u$-channel for $\lambda^-_\sigma$, i.e.,

$$\text{Flow}^{(u)}_{(\lambda^-_\sigma)}\left(-\bar{p} - P/2, \bar{p} - P/2, -\bar{p}' + P/2, \bar{p}' + P/2\right) = -\text{Flow}^{(t)}_{(\lambda^-_\sigma)}\Big|_{q_t \to q_u, t_1 \to u_1, t_2 \to u_2}, \quad \text{(B.28)}$$

where the minus sign on the right side arises from the symmetry relation for the antisymmetric four-quark dressings in (A.11). The flow of $s$-channel for the $\lambda_\sigma^-$ is given by

$$
\begin{aligned}
&\text{Flow}_{(\lambda_\sigma^-)}^{(s)}\left(-\bar{p}-P/2, \bar{p}-P/2, -\bar{p}'+P/2, \bar{p}'+P/2\right)\\
&= \int \frac{d^4 q}{(2\pi)^4} \frac{\bar{t}_s(q)}{Z_q(q)Z_q(q_s)} \bar{G}_s(q_s) \left\{-2q^2\left[1+r_q\left(\frac{q^2}{k^2}\right)\right] M_q(q)M_q(q_s)\left(3\lambda_\pi^-(s_1)\lambda_\pi^-(s_2)+\lambda_\sigma^-(s_1)\lambda_\sigma^-(s_2)\right)\right.\\
&\left.+\frac{q\cdot q_s}{12}\left[1+r_q\left(\frac{q_s^2}{k^2}\right)\right]\left[M_q^2(q)-q^2\left[1+r_q\left(\frac{q^2}{k^2}\right)\right]^2\right]\left(\lambda_\sigma^-(s_1)\lambda_\pi^-(s_2)+\lambda_\pi^-(s_1)\lambda_\sigma^-(s_2)-2\lambda_\pi^-(s_1)\lambda_\pi^+(s_2)\right)\right\}.
\end{aligned}
\tag{B.29}
$$

The flow of $t$-channel for the $\lambda_\pi^-$ reads

$$
\begin{aligned}
&\text{Flow}_{(\lambda_\pi^-)}^{(t)}\left(-\bar{p}-P/2, \bar{p}-P/2, -\bar{p}'+P/2, \bar{p}'+P/2\right)\\
&= \int \frac{d^4 q}{(2\pi)^4} \frac{\bar{t}_s(q)}{Z_q(q)Z_q(q_t)} \bar{G}_s(q_t) \left\{-2q^2\left[1+r_q\left(\frac{q^2}{k^2}\right)\right] M_q(q)M_q(q_t)\, \mathcal{A}_{(\lambda_\pi^-)}^{(t)}(t_1,t_2)\right.\\
&\left.+\frac{q\cdot q_t}{24}\left[1+r_q\left(\frac{q_t^2}{k^2}\right)\right]\left[M_q^2(q)-q^2\left[1+r_q\left(\frac{q^2}{k^2}\right)\right]^2\right]\mathcal{B}_{(\lambda_\pi^-)}^{(t)}(t_1,t_2)\right\},
\end{aligned}
\tag{B.30}
$$

with

$$
\begin{aligned}
\mathcal{A}_{(\lambda_\pi^-)}^{(t)}(t_1,t_2) =&\ 13\lambda_\pi^+(t_1)\lambda_\pi^+(t_2)-\lambda_\sigma^-(t_1)\lambda_\pi^-(t_2)+12\lambda_\pi^+(t_1)\lambda_\pi^-(t_2)+\lambda_\sigma^+(t_1)\lambda_\pi^-(t_2)\\
&+11\lambda_\pi^-(t_1)\lambda_\pi^-(t_2)-\lambda_\pi^-(t_1)\lambda_\sigma^-(t_2)+12\lambda_\pi^-(t_1)\lambda_\pi^+(t_2)+\lambda_\pi^-(t_1)\lambda_\sigma^+(t_2),
\end{aligned}
\tag{B.31}
$$

and

$$
\begin{aligned}
\mathcal{B}_{(\lambda_\pi^-)}^{(t)}(t_1,t_2) =&\ 310\lambda_\pi^+(t_1)\lambda_\pi^+(t_2)-13\lambda_\pi^+(t_1)\lambda_\sigma^-(t_2)-13\lambda_\sigma^-(t_1)\lambda_\pi^+(t_2)+13\lambda_\sigma^+(t_1)\lambda_\pi^+(t_2)\\
&-11\lambda_\sigma^-(t_1)\lambda_\pi^-(t_2)+290\lambda_\pi^+(t_1)\lambda_\pi^-(t_2)+11\lambda_\sigma^+(t_1)\lambda_\pi^-(t_2)+13\lambda_\pi^+(t_1)\lambda_\sigma^+(t_2)\\
&+262\lambda_\pi^-(t_1)\lambda_\pi^-(t_2)-11\lambda_\pi^-(t_1)\lambda_\sigma^-(t_2)+290\lambda_\pi^-(t_1)\lambda_\pi^+(t_2)\\
&+11\lambda_\pi^-(t_1)\lambda_\sigma^+(t_2).
\end{aligned}
\tag{B.32}
$$

Similar with (B.28), one arrives at

$$
\text{Flow}_{(\lambda_\pi^-)}^{(u)}\left(-\bar{p}-P/2, \bar{p}-P/2, -\bar{p}'+P/2, \bar{p}'+P/2\right) = -\text{Flow}_{(\lambda_\pi^-)}^{(t)}\bigg|_{q_t\to q_u,\, t_1\to u_1,\, t_2\to u_2}.
\tag{B.33}
$$

The flow of $s$-channel for the $\lambda_\pi^-$ reads

$$
\begin{aligned}
&\text{Flow}_{(\lambda_\pi^-)}^{(s)}\left(-\bar{p}-P/2, \bar{p}-P/2, -\bar{p}'+P/2, \bar{p}'+P/2\right)\\
&= \int \frac{d^4 q}{(2\pi)^4} \frac{\bar{t}_s(q)}{Z_q(q)Z_q(q_s)} \bar{G}_s(q_s) \left\{-2q^2\left[1+r_q\left(\frac{q^2}{k^2}\right)\right] M_q(q)M_q(q_s)\right.\\
&\times\left(\lambda_\pi^-(s_1)\lambda_\pi^-(s_2)+\lambda_\sigma^-(s_1)\lambda_\sigma^-(s_2)\right)+\frac{q\cdot q_s}{12}\left[1+r_q\left(\frac{q_s^2}{k^2}\right)\right]\left[M_q^2(q)-q^2\left[1+r_q\left(\frac{q^2}{k^2}\right)\right]^2\right]\\
&\left.\times\left(2\lambda_\pi^-(s_1)\lambda_\pi^-(s_2)-\lambda_\sigma^-(s_1)\lambda_\pi^-(s_2)-\lambda_\pi^-(s_1)\lambda_\sigma^-(s_2)\right)\right\}.
\end{aligned}
\tag{B.34}
$$

# C Flow equations of the quark propagator

We flow equations for the quark mass function and quark wave function is readily computed from that of the quark self-energy shown in Figure 14. The flow of the quark mass function

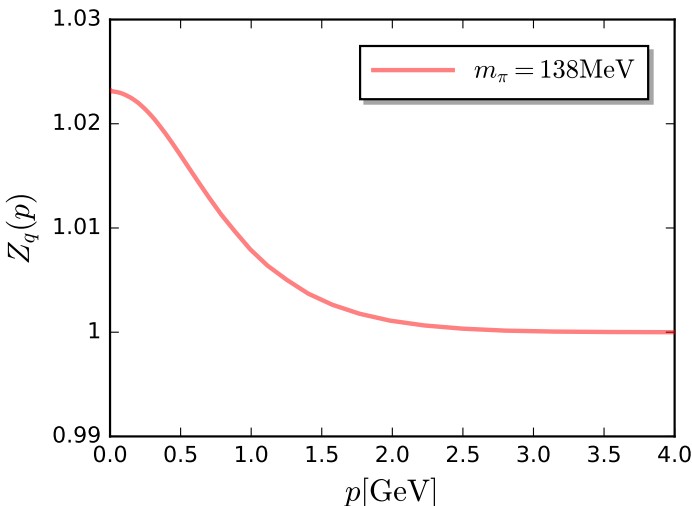

Figure 17: Quark wave function as a function of the momentum at pion mass $m_\pi = 138$ MeV with the initial cutoff scale $\Lambda = 850$ MeV.

is obtained by tracing the flow of the self-energy, $\partial_t \Gamma^{(2)}_{q\bar{q}}(p)$, in Dirac and flavour space, also using the isospin symmetry: tr $\partial_t \Gamma^{(2)}_{q\bar{q}}(p)$. With an appropriate normalisation this leads us to

$$\partial_t M_q(p) = \eta_q(p) M_q(p) + \int \frac{d^4 q}{(2\pi)^4} \frac{\bar{t}_s(q)}{Z_q(p)Z_q(q)} M_q(q) \left[1 + r_q\left(\frac{q^2}{k^2}\right)\right] \left\{-3\lambda_\pi^+ - 23\lambda_\sigma^+ + 3\lambda_a^+ - \lambda_\eta^+ - \frac{16}{3}\lambda_{(S+P)_-^{\mathrm{adj}}}^+ \right.$$

$$\left. + \frac{32}{3}\lambda_{(S+P)_+^{\mathrm{adj}}}^+ + 8\lambda_{(V+A)}^+ + 3\lambda_\pi^- - 25\lambda_\sigma^- - 3\lambda_a^- + \lambda_\eta^- + \frac{16}{3}\lambda_{(S+P)_-^{\mathrm{adj}}}^- - \frac{32}{3}\lambda_{(S+P)_+^{\mathrm{adj}}}^- - 8\lambda_{(V+A)}^- \right\}. \quad \text{(C.1)}$$

The momenta of the four-quark dressings, which are not shown explicitly in (C.1), are given by

$$\lambda_a(-p, p, -q, q) = \lambda_a\left(s = (p+q)^2, t = 0, u = (p-q)^2\right), \quad \text{(C.2)}$$

where the three-momentum-channel truncation for the four-quark vertices in (21) has been employed. The flow equation of the quark wave function is given by

$$\partial_t Z_q(p) = \int \frac{d^4 q}{(2\pi)^4} \frac{\bar{t}_s(q)}{Z_q(q)} \frac{p \cdot q}{p^2} \left[M_q^2(q) - q^2\left[1 + r_q\left(\frac{q^2}{k^2}\right)\right]^2\right] \left\{\frac{3}{2}\lambda_\pi^+ + \frac{1}{2}\lambda_\sigma^+ - \frac{3}{2}\lambda_a^+ - \frac{1}{2}\lambda_\eta^+ - \frac{8}{3}\lambda_{(S-P)_-^{\mathrm{adj}}}^+ - 10\lambda_{(V-A)}^- \right.$$

$$\left. - 12\lambda_{(V+A)}^+ - 14\lambda_{(V-A)}^+ - \frac{8}{3}\lambda_{(V-A)^{\mathrm{adj}}}^+ - \frac{3}{2}\lambda_\pi^- - \frac{1}{2}\lambda_\sigma^- + \frac{3}{2}\lambda_a^- + \frac{1}{2}\lambda_\eta^- + \frac{8}{3}\lambda_{(S-P)_-^{\mathrm{adj}}}^- - 12\lambda_{(V+A)}^- + \frac{8}{3}\lambda_{(V-A)^{\mathrm{adj}}}^- \right\}. \quad \text{(C.3)}$$

In Figure 17 we show the quark wave function, which only deviates slightly from unity in the regime of small momentum. The complete calculation of quark wave functions will be done in QCD.

## D    Error estimate for $s, t, u$-channel approximation

In Section 2.2 we have discussed the $s, t, u$-channel approximation summarised in (22), which we repeat here for the sake of convenience:

$$\lambda_\alpha^\pm(p_1, p_2, p_3, p_4) = \lambda_\alpha^\pm(s, t, u) + \Delta\lambda_\alpha^\pm(p_1, p_2, p_3, p_4), \quad \text{(D.1a)}$$

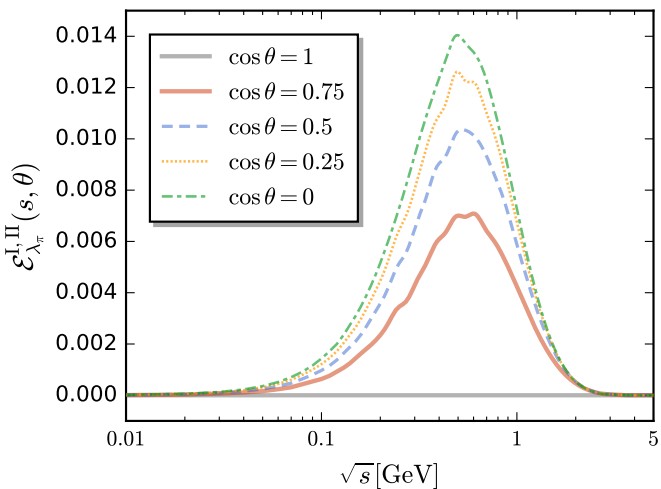

Figure 18: Relative error of the $s, t, u$-channel approximation of the four-quark dressing in the $\pi$ channel, $\mathcal{E}^{\mathrm{I,II}}_{\lambda_\pi}(s, \theta)$, (D.6), as functions of $\sqrt{s}$ for different values of $\cos\theta$ in the momentum configuration II, (D.3).

where the $s, t, u$-channel parts $\lambda^\pm_\alpha(s, t, u)$ are computed from

$$P_\mu = \sqrt{P^2}\left(1, 0, 0, 0\right), \quad \bar{p}_\mu = \sqrt{p^2}\left(1, 0, 0, 0\right), \quad \bar{p}'_\mu = \sqrt{p^2}\left(\cos\theta, \sin\theta, 0, 0\right), \quad \text{(D.1b)}$$

which defines a three-dimensional subspace of momentum configurations, that are in one-to-one correspondence with the Mandelstam momenta

$$s = 2p^2(1 + \cos\theta), \qquad t = P^2, \qquad u = 2p^2(1 - \cos\theta). \quad \text{(D.1c)}$$

Finally we approximate

$$\Delta\lambda^\pm_\alpha(p_1, p_2, p_3, p_4) \approx 0, \quad \text{(D.1d)}$$

which reduces the six momentum and angular variables in the dressings $\lambda^\pm_\alpha(p_1, p_2, p_3, p_4)$ to three variables.

Equation (D.1d) allows for the following self-consistency check: We assume $\Delta\lambda^\pm_\alpha \approx 0$ on the right hand side of the flow equation. Then, $\Delta\lambda^\pm_\alpha$ can readily be obtained from its integrated flow, computed within the approximation (D.1). A measure of its importance is the relative contribution in comparison to the $s, t, u$-channel part,

$$\mathcal{E}_{\lambda^\pm_\alpha}(\boldsymbol{p}) = \left|\frac{\lambda^\pm_\alpha(\boldsymbol{p}) - \lambda^\pm_\alpha(s, t, u)}{\lambda^\pm_\alpha(\boldsymbol{p})}\right| \approx \left|\frac{\Delta\lambda^\pm_\alpha(\boldsymbol{p})}{\lambda^\pm_\alpha(s, t, u)}\right|, \quad \text{(D.2)}$$

where we finally dropped terms of the order $\Delta\lambda^\pm_\alpha$ on the right hand side.

For the evaluation of (D.2) we define another three-dimensional subspace of momentum variables that differs from (D.1b),

$$P_\mu = \sqrt{P^2}\left(1, 0, 0, 0\right), \quad \bar{p}_\mu = \sqrt{p^2}\left(\cos\theta, \sin\theta, 0, 0\right), \quad \bar{p}'_\mu = \sqrt{p^2}\left(-\sin\theta, \cos\theta, 0, 0\right), \quad \text{(D.3)}$$

with $\theta \in [0, \pi]$ and the Mandelstam variables

$$s = u = 2p^2, \qquad t = P^2. \quad \text{(D.4)}$$

Evidently, the three-dimensional subspace (D.3) picks out a two-dimensional subspace in $\{s, t, u\}$.

We compute the relative error (D.2) for the largest (and only relevant) four-quark dressing, the pseudo-scalar dressing $\lambda_\pi$. It can be computed in two distinct ways: First we can evaluate the radial momentum dependence for a fixed relative size of $s, t, u$-channel momenta. Here we choose

$$s = t = u, \qquad \longrightarrow P_I^2 = 2p_I^2, \quad \cos\theta_I = 0, \qquad \text{and} \qquad P_{II}^2 = 2p_{II}^2, \tag{D.5}$$

obtained from either the momentum configuration (D.1b) (I) or (D.3) (II). While the cosine $\cos\theta_I = 0$ is fixed, the cosine $\cos\theta_{II}$ can take any value in [-1,1]. The respective dressings are $\lambda_\pi^I(s)$ and $\lambda_\pi^I(s,\theta)$ and hence the relative error (D.2) is a function of $\sqrt{s}$ and $\cos\theta$,

$$\mathcal{E}_{\lambda_\pi}^{I,II}(s,\theta) = \frac{|\lambda_\pi^I(s) - \lambda_\pi^{II}(s,\theta)|}{|\lambda_\pi^I(s)|}. \tag{D.6}$$

In Figure 18 we show the relative error (D.6) as functions of $\sqrt{s}$ for different values of $\cos\theta$. This provides us with an error estimate for the approximation (D.1d), which is smaller than 1.5%. This indicates that the three-momentum-channel approximation of the four-quark vertices discussed in Section 2.2 is of quantitative reliability.

Moreover, we replace momentum configuration II in (D.3) with another one, denoted by III as follows

$$P_\mu = \sqrt{P^2}\Big(1, 0, 0, 0\Big), \quad \bar{p}_\mu = \sqrt{p^2}\Big(\cos\theta, \sin\theta, 0, 0\Big), \quad \bar{p}'_\mu = \sqrt{p^2}\Big(\cos\theta, \sin\theta, 0, 0\Big), \tag{D.7}$$

with $\theta \in [0, \pi]$, and compute

$$\mathcal{E}_{\lambda_\pi}^{I,III}(s,\theta) = \frac{|\lambda_\pi^I(s) - \lambda_\pi^{III}(s,\theta)|}{|\lambda_\pi^I(s)|}, \tag{D.8}$$

with $t = 0$ and $u = 0$. The relevant results are shown in Figure 19. The error of the configuration choice is more than one order of magnitude smaller than that depicted in Figure 18.

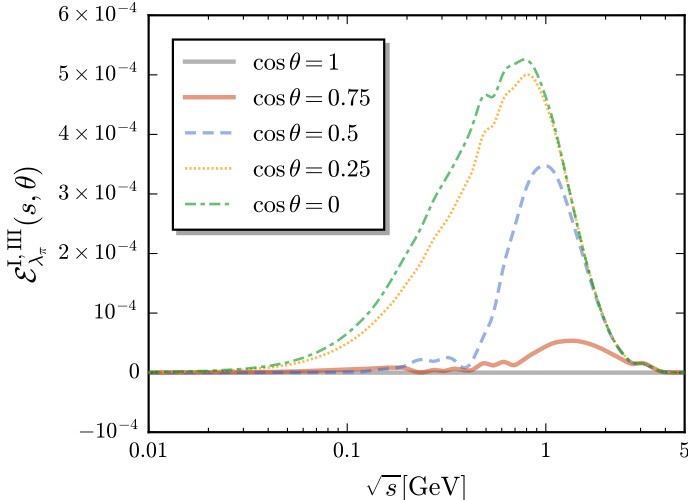

Figure 19: Relative error of the $s, t, u$-channel approximation of the four-quark dressing in the $\pi$ channel, $\mathcal{E}_{\lambda_\pi}^{I,III}(s,\theta)$, (D.8), as functions of $\sqrt{s}$ for different values of $\cos\theta$ in the momentum configuration III, (D.7).

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
