# Peer review of "Four-quark scatterings in QCD II"

_SciPost Physics, doi:SciPost Phys. 17, 148 (2024)_

## Round 2 · Referee Report · Anonymous (Referee 1) · 2024-6-19

Strengths

1-The findings are important, in particular the identification of a reliable approximation scheme for the fRG computation of the four-quark vertex, and the scale below which the effective theory reproduces QCD;
2-The approximations employed, such as restricting the dressing functions' kinematic dependence to the s, t, u channels, are clearly stated and well-motivated;
3-The errors introduced by these approximations are quantified, as much as possible;
4-Ways forward are identified for follow-up research.
5-The article is well structured;

Report

The manuscript details an approximation scheme for the computation of the four-quark vertex through the Functional Renormalization Group (fRG), through which dynamical chiral symmetry breaking and the formation of bound states can be studied quantitatively. The approximation scheme relies on three main steps: (i) dynamical hadronization, where effective fields are introduced to describe the interaction of composite states (sigma and pion, in the present case); (ii) the identification of a consistent subset of tensor structures for the vertex, which respects the symmetries of the theory and are expected to be dominant at low energies; (iii) the simplification of the kinematic dependence of the attending dressing functions to functions of the s, t, and u variables only, rather than the six variables allowed by Lorentz symmetry. The authors find that the t channel alone is not sufficient for quantitative reliability, but that the three channels s, t and u, suffice. Moreover, they demonstrate that the sigma-pion tensor structures dominate the infrared dynamics, and identify the cutoff scale, namely 500 MeV, beyond which QCD fails to be reproduced quantitatively, indicating that above this scale the inclusion of the gluon degrees of freedom is necessary.
These conclusions are firmly supported by explicitly estimating the errors introduced by these approximations through a series of numerical experiments.
These results are certainly an important step towards a full QCD description of the dynamical chiral symmetry breaking and bound state formation in the fRG framework. Moreover, the manuscript is well structured, and the presentation is quite clear. I thus recommend its publication in present form.

Recommendation

Publish (easily meets expectations and criteria for this Journal; among top 50%)

---

## Round 2 · Referee Report · Anonymous (Referee 2) · 2024-7-6

Strengths

1) This work is part of a series of articles aimed at describing the emergence of mesonic bound states from the infrared dynamics of four-quark scatterings in QCD, employing the functional renormalisation group (fRG).

2) The manuscript is solid, the organization of the article is good, and the presentation of the material is coherent.

3) The results presented here set the stage for the final step towards quantitative predictions of the QCD resonance structure and general timelike correlation functions in first-principles functional QCD using the fRG.

Report

The present manuscript is part of a series of articles in which the authors study four-quark scatterings in QCD, employing the functional renormalisation group (fRG) approach. More specifically, they have computed the four-quark dressings, including the s, t, and u channels. They find that, in this improved approximation, the four-quark vertex can be accurately estimated. In particular, they computed the pion pole mass, pion decay constant, Bethe-Salpeter amplitudes, and the quark mass and wave functions. The general idea and the results presented in the manuscript are quite interesting. As the authors emphasized, this work sets the stage for the final step towards quantitative predictions of the QCD resonance structure and general timelike correlation functions in first-principles functional QCD using the fRG.

From a technical standpoint, the paper is solid, the organization of the article is good, and the presentation of the material is coherent. For these reasons, the article merits publication in SciPost. However, before that, the authors might consider providing some additional details/discussion about the following points in the paper.

1) In Fig. 7, the authors compare their result for the dynamical quark mass with the lattice result of Ref. [37], which was obtained with a pion mass of 280 MeV. For this particular case, the current quark mass employed in that lattice simulation was $m_q=6.2$ MeV, whereas the authors fixed their current mass at $m_q=19.2$ MeV. As we know, the dynamical quark mass displays considerable dependence on the values of the current masses used as input. Could the authors comment on these differences in the current masses employed in this comparison, and how this would affect the agreement between the fRG and lattice results presented in this plot?

2) In the same Ref. [37], there are lattice results for the quark wave function, which is also computed in the present manuscript and shown in Fig. 17. However, the authors did not compare their prediction for this function with the lattice results. Could the authors comment on this? What are the main differences between both sets of results?

Requested changes

I have no additional modification requests, apart from the clarification points mentioned above

Recommendation

Publish (easily meets expectations and criteria for this Journal; among top 50%)

  • validity: top
  • significance: high
  • originality: top
  • clarity: top
  • formatting: perfect
  • grammar: excellent

Author:  Chuang Huang  on 2024-10-10  [id 4854]

(in reply to Report 2 on 2024-07-06)
Category:
answer to question

We are grateful to the referee for the positive comments on this manuscript. Here are our responses to the questions raised by the referee. We have also added relevant discussions after Eq.(37) on page 7 and Eq.(C3) on page 17 of the paper.

The referee writes:

1) In Fig. 7, the authors compare their result for the dynamical quark mass with the lattice result of Ref. [37], which was obtained with a pion mass of 280 MeV. For this particular case, the current quark mass employed in that lattice simulation was $m_{q}=6.2$ MeV, whereas the authors fixed their current mass at $m_{q}=19.2$ MeV. As we know, the dynamical quark mass displays considerable dependence on the values of the current masses used as input. Could the authors comment on these differences in the current masses employed in this comparison, and how this would affect the agreement between the fRG and lattice results presented in this plot?

Our response: For the first comment, we think there are two points that need clarification. Firstly, this work is done within the low-energy effective theory of four-quark scattering, without considering the glue dynamics. Correspondingly, the cutoff scale $\Lambda=500\,\mathrm{MeV}$ is also lower than the QCD cutoff scale $\Lambda_{\mathrm{QCD}}\sim 2 -5\,\mathrm{GeV}$. This is also why the initial quark masses are relatively large at the cutoff scale. In the third step paper, we will obtain quark masses from first-principles QCD and conduct a more precise comparison with lattice results. Secondly, in reference [37], $m_{q}=6.2\,\text{MeV}$ is the subtracted bare quark mass under the $\overline{\mathrm{MS}}$ renormalization scheme, while the fRG method corresponds to $\mathrm{MOM}^{2}$ scheme in the ultraviolet region, see in reference [39]. Therefore, the current quark mass at the cutoff scale is not a physical quantity that can be directly compared between the fRG method and the lattice method. In different approaches, only physical quantities that are almost independent of the renormalization scheme can be directly compared, such as momentum-dependent quark mass function at the IR scale, meson masses, meson decay constants, and so on.

The referee writes:

2) In the same Ref. [37], there are lattice results for the quark wave function, which is also computed in the present manuscript and shown in Fig. 17. However, the authors did not compare their prediction for this function with the lattice results. Could the authors comment on this? What are the main differences between both sets of results?

Our response: Through simple analysis of the Dirac structure ($\gamma$ matrix) of the quark two-point functions in QCD flow equations, it can be inferred qualitatively that the quark wave function is primarily dominated by the quark self-energy diagrams which include the quark-gluon vertices. The four-quark vertices studied in this work mainly contribute to generating quark masses. In Fig.17 it is illustrated that even when considering the $s,t,u$-channel momentum dependence, the quark wave function resulting from the four-quark interactions is still away from the unity very mildly, which confirms the simple analysis above quantitatively. Furthermore, it is worth mentioning that currently there are qualitative differences in the results of quark wave functions obtained by different lattice QCD collaboration groups, see in reference [37] and Phys.Rev.D 104 (2021) 9, 094509. Obviously, further studies of quark wave functions are required in QCD calculation. In the third paper in this series, we will calculate the quark wave functions from the first-principles QCD within the fRG method.

---

## Editorial Decision

published